# Taming large-scale genomic analyses via sparsified genomics

Mohammed Alser [1,2,3] ✉, Julien Eudine [1] & Onur Mutlu[1]

Searching for similar genomic sequences is an essential and fundamental step in biomedical research. State-of-the-art computational methods performing such comparisons fail to cope with the exponential growth of genomic sequencing data. We introduce the concept of sparsified genomics where we systematically exclude a large number of bases from genomic sequences and enable faster and memory-efficient processing of the sparsified, shorter genomic sequences, while providing comparable accuracy to processing non-sparsified sequences. Sparsified genomics provides benefits to many genomic analyses and has broad applicability. Sparsifying genomic sequences accelerates the state-of-the-art read mapper (minimap2) by 2.57-5.38x, 1.13-2.78x, and 3.52-6.28x using real Illumina, HiFi, and ONT reads, respectively, while providing comparable memory footprint, 2x smaller index size, and more correctly detected variations compared to minimap2. Sparsifying genomic sequences makes containment search through very large genomes and large databases 72.7-75.88x (1.62-1.9x when indexing is preprocessed) faster and 723.3x more storage-efficient than searching through non-sparsified genomic sequences (with CMash and KMC3). Sparsifying genomic sequences enables robust microbiome discovery by providing 54.15-61.88x (1.58-1.71x when indexing is preprocessed) faster and 720x more storage-efficient taxonomic profiling of metagenomic samples over the state-of-the-art tool (Metalign).

Global efforts are underway to identify the earth's virome in preparation for the next pandemic and to better study genomic diversity by building population-specific representative genomes[1–4]. These challenging targets are now within reach thanks to modern high-throughput sequencing (HTS) technologies that quickly decrypt the nucleotide sequences of an individual's DNA. The output of HTS systems consists of sets of genomic sequences, usually referred to as reads, extracted randomly from an individual's genome sequence. The length of each read is orders of magnitude smaller than the complete genome sequence, ranging from a few hundred to a few million base pairs (bp). Contemporary HTS technologies are capable of generating tens of millions to billions of reads per sample, and the throughput of prominent systems from Illumina[5], Pacific Biosciences (PacBio)[6,7], and Oxford Nanopore Technologies (ONT)[8,9] is constantly being improved. Recent years

have seen a surge in genome sequencing projects, resulting in tens of petabases (e.g., $18.2 \times 10^{15}$ bases in GenBank alone) of publicly available sequencing data. We are witnessing an increase in the number of genome assemblies for different organisms and populations[3,10–13], which will continue to rise with initiatives such as the Human Pangenome Reference Consortium. The genomic data that needs to be exhaustively searched and analyzed will quickly grow with such efforts and the existence of ever-more-powerful sequencing technologies.

Most genomic analyses taking advantage of the tsunami of sequencing data include computational steps for searching and analyzing genomic sequences for different purposes and goals. A few of these goals can be discovering genomic variations and disease causes in clinical medicine[14–17], identifying complex genomic rearrangements in cancer[18], rapid surveillance of disease outbreaks[19,20], and

[1]Department of Information Technology and Electrical Engineering, ETH Zürich, Zurich, Switzerland. [2]Department of Computer Science, Georgia State University, Atlanta, GA, USA. [3]Department of Clinical Pharmacy, University of Southern California, LA, CA, USA. ✉e-mail: mealser@gmail.com

understanding pathogens and urban microbial communities[2,21,22]. Computational genomic analyses performing such comparisons start with obtaining genomic data by sequencing new samples or downloading real data from publicly available databases (Fig. 1). The analyses then employ indexing and seeding techniques to quickly locate subject data without having to sequentially search every genomic nucleotide in one or more very long genome sequences for each sequencing read. Indexing for searching genomic sequences was first used in 1988 by FASTA[23] and has since dominated the landscape of genomic tools[24]. Data indexes store short subsequences, called seeds, extracted from the target genome (or database of genomic sequences) along with their start locations. Seeding usually performs a very similar algorithm to indexing, but seeding does not store the resulting seeds. Instead, the seeding step extracts seeds from the query sequences and examines their existence in the index. Matching a genomic sequence to one or more genomic sequences requires matching seeds extracted from the query sequence to seeds stored in the index, which reduces the search space from the entire target genome sequence to only the neighborhood regions of each seed location.

State-of-the-art computational methods analyzing genomic sequences fail to cope with the exponential growth of genomic sequencing data. Despite the benefits of indexing and seeding, they can still drastically degrade the overall performance, memory footprint, storage footprint, and accuracy of genomic analyses depending on: (1) how fast indexing and seeding can calculate the resulting seeds, (2) how fast indexing can insert seeds to the index, (3) how fast seeding can query the index, (4) the type and length of resulting seeds, and (5) the size of the generated index[24–36]. Many attempts were made to facilitate searching large genomic data and finding similar genomic sequences (Supplementary Note 1). Recent attempts tend to follow one of three key directions: (1) Building smaller data indexes for faster index access and traversal by extracting a smaller number of seeds from genomic sequences[37–42], (2) Reducing indexing and seeding overhead by avoiding the use of computationally expensive seeds[25,27,35,43–45], and (3) Alleviating the accuracy degradation that results from considering only exactly-matching seeds between two sequences by using sparse seeds (e.g., spaced seeds) or variable-length seeds[46–60]. To our knowledge, most state-of-the-art computational methods suffer from four critical limitations.

(1) Existing analysis tools usually consider the indexing step as a *preprocessing* step (i.e., the indexing time is not included in the analysis time), and thus, there has been little to no effort invested in improving the execution time of indexing. The indexing step can be performed only once as a preprocessing step for each different genome version, as done in read mapping[24,32]. However, the indexing step can also be performed online, i.e., during analysis, where indexing time contributes to the total analysis time. Examples include taxonomic profiling of metagenomic samples[21,22], matching statistics[56,61], identifying de novo variations by comparing sequencing reads of family members[62], and identifying somatic variations by comparing reads sequenced from both healthy and tumor cells of the same patient[18]. Even in read mapping, there is a critical need for mapping reads to different genome assemblies of the same organism[63,64], where the index needs to be re-built for each reference genome version. For example, we observe that the size of NCBI RefSeq, the widely used publicly available database for reference genomes, doubles nearly every 3 years, totaling >3 Tbp of sequencing data as of July 2022, while its number of distinct organisms doubles every 6 years (Supplementary Note 2). This indicates that RefSeq includes different genome assemblies for the same organism more often than assemblies for new organisms, which needs to be analyzed and studied.

(2) Regardless of the seed type used, state-of-the-art indexing and seeding methods calculate all possible overlapping seeds and their hash values before deciding on which seed to consider for indexing and seeding and which seed to exclude. The same implementation used for indexing is usually used also for the seeding step to ensure applying the same seed extraction technique to the indexed sequence and query sequence. This worsens the problem as the seeding step contributes to the analysis time and the indexing and seeding computations are currently performed sequentially. For example, minimap2[37] (a state-of-the-art, commonly used, and well-maintained read mapper) and its recent optimized implementation[33] are still using a non-vectorized/non-parallelized implementation of both indexing and seeding (i.e., `sketch.c` in 2.24-r1122 version of minimap2 as of 11 November 2022). The contribution of both the indexing and seeding steps to the total execution time of genomic analysis is different from one analysis to another. For example, the execution time of both the indexing and seeding steps accounts for ~10–27% of the total read mapping time and 97% of the total time for taxonomic profiling of metagenomic samples.

(3) The state-of-the-art indexing and seeding methods over-represent genomic sequences with redundant information. There are

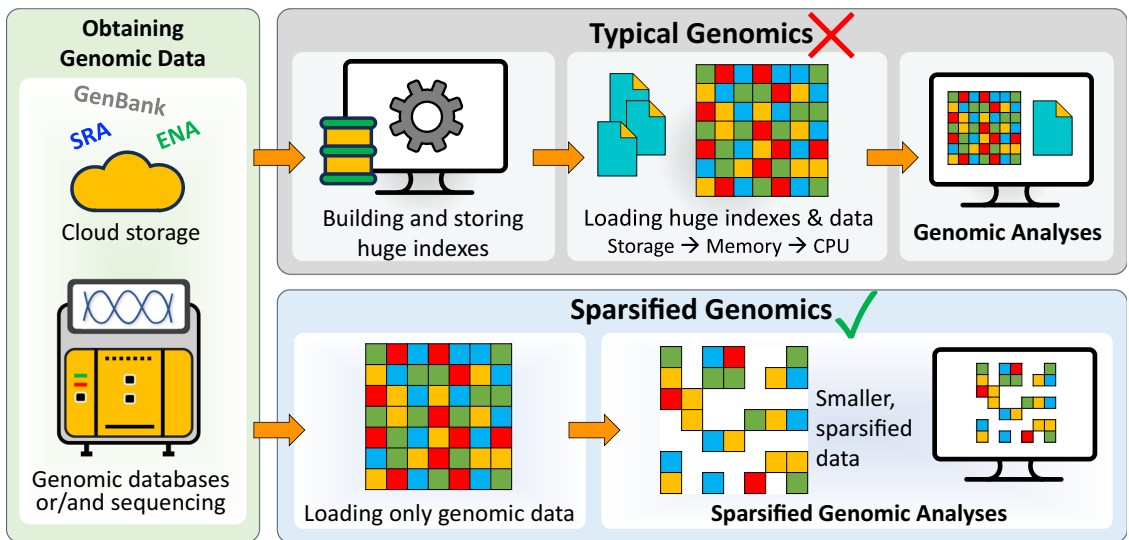

**Fig. 1 | Overview of typical genomic analyses versus the proposed sparsified genomic analyses.** Both approaches start with obtaining genomic data through sequencing a new sample, downloading from publicly available databases, or both. Unlike all traditional genomic analyses, sparsified genomics shortens genomic sequences before performing computations.

at least two examples of such redundant information. (1) Each base in a genomic sequence can appear in multiple overlapping seeds, causing additional overhead and repeated computation[65]. (2) The number of seeds extracted from each read sequence can be excessively large as it is proportional to the read length. Reducing the redundant information (e.g., using a larger $w$ parameter, which determines the number of overlapping seeds that can be represented by a single seed, value in minimap2[37]) reduces the index size and the number of seed lookups in the index, but also drastically reduces the accuracy of the analysis. The accuracy in this context is defined as the sensitivity, i.e., the number of query sequences that are correctly matched to one or more candidate regions in the target genomic sequences divided by the total number of input query sequences.

(4) Even after excluding a large number of seeds, the generated index is normally tremendously large[24–31]. The size of the index can be up to 21.25× larger than the size of the indexed genome, assuming the indexed genome is 2-bit encoded (Supplementary Table 1). Generating large indexes precludes easy sharing across networks, which limits both the portability of analysis and the reproducibility of results[66]. Processing large indexes requires using a very powerful computing infrastructure (with very expensive large main memory and tens of thousands of CPUs[1]), which is usually available only at a limited number of places. We now need more than ever to catalyze and accelerate genomic analyses by addressing the four critical limitations. Our goal is to enable ultra-fast and highly efficient indexing and seeding steps in various genomic analyses so that pre-building genome indexes for each genome assembly is *no* longer a requirement for quickly running large-scale genomic analyses using large genomes and various versions of genome assembly.

In this work, we introduce the concept of sparsified genomics (Fig. 1). The key idea is to systematically exclude a large number of bases from genomic sequences and enable the processing of the sparsified, shorter genomic sequence while maintaining similar or higher accuracy compared to that of processing non-sparsified sequences. We exploit redundancy in genomic sequences to eliminate some regions in the genomic sequences to reduce the input workload of each step of genomic analysis and accelerate overall performance. To demonstrate the benefits of sparsified genomics in real genomic applications, we introduce Genome-on-Diet, the first highly parallel, memory-frugal, and accurate framework for sparsifying genomic sequences (Fig. 2). Genome-on-Diet is based on four key ideas. (1) Using a repeating pattern sequence to decide which base(s) in the genomic sequence should be excluded and which base(s) should be included. The pattern sequence is a user-defined, fully configurable shortest repeating substring that represents included and excluded bases via 1's and 0's, respectively. Genome-on-Diet manipulates only a copy of the genomic sequence that is used for the initial steps of an analysis, such as indexing and seeding. The original genomic sequence is still maintained (by default) for performing accuracy-critical steps of an analysis, if needed, such as base-level sequence alignment where all bases must be accounted for. Users can disable maintaining the original genomic sequence by using the --idx-no-seq parameter setting. (2) Inferring at which location of the query sequence the pattern should be applied in order to correctly match the included bases of the query sequence with the included bases of the target sequences. Applying the pattern always starting from the first base of the read sequence can lead to poor results due to a possible lack of seed

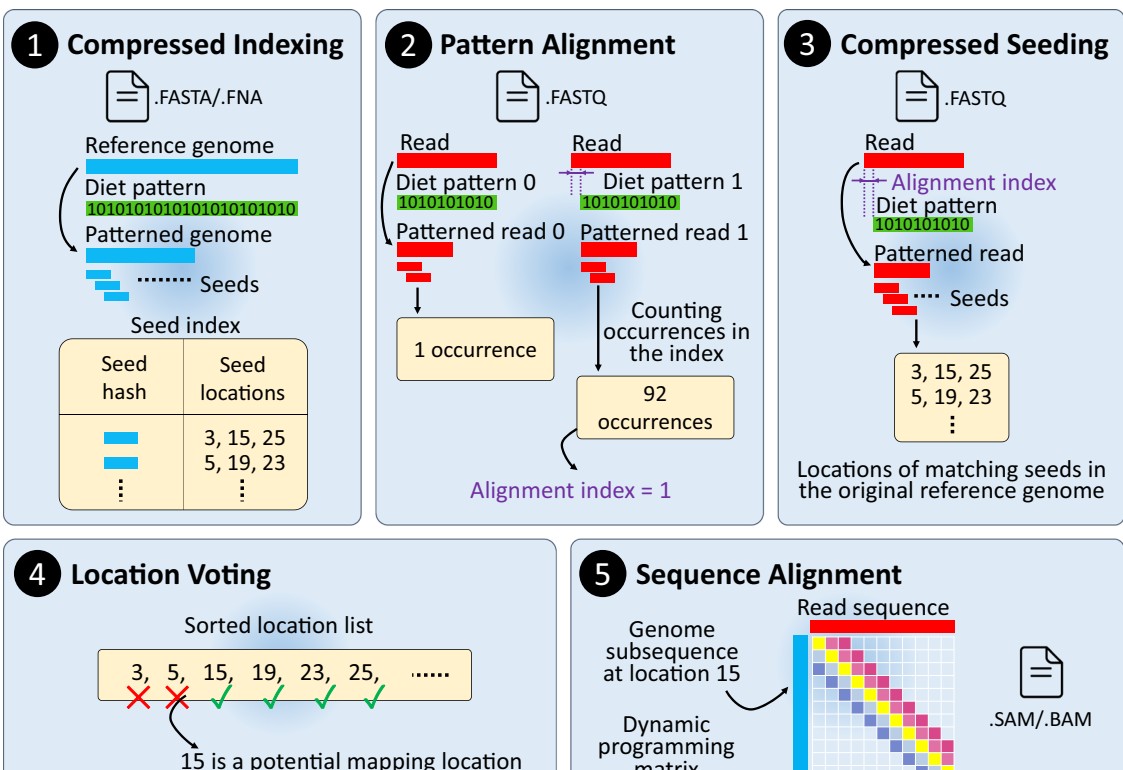

**Fig. 2 | Overview of the five main computational steps of the Genome-on-Diet framework for enabling sparsified data processing.** We explain the five steps in the context of read mapping. ❶ A user-defined pattern, called diet pattern, is applied to the reference genome to obtain a shorter 'compressed' version, called patterned genome, of the genome sequence. The seeds collected from the patterned genome are extracted and stored in an index structure, called seed index, along with their start locations in the reference genome. ❷ Multiple shifted versions of the diet pattern are applied to each read sequence to calculate the alignment index based on which pattern version leads to the highest total number of seed occurrences. ❸ The diet pattern is shifted by the alignment index value and applied to the read sequence. The resulting patterned read is used to extract seeds and query the seed index. ❹ Location voting examines the adjacent locations of seed matches and quickly decides whether or not the computationally expensive sequence alignment is needed. ❺ Sequence alignment is calculated only for sequence pairs that are accepted by location voting to generate the alignment file (e.g., in SAM format).

matches. (3) Providing a highly parallel and highly optimized implementation of indexing and seeding using modern single-instruction multiple-data (SIMD) instructions employed in modern microprocessors[67]. (4) Introducing four key optimization strategies to enable high parallelism, efficiency, and accuracy.

Sparsified genomics provides in principle several key benefits: (1) Reduced total execution time due to processing less workload (smaller number of included bases), (2) Reduced peak memory footprint due to a smaller number of extracted seeds and hence a smaller index, (3) No need to pre-build genome indexes as it is feasible to build it during the analysis with low performance-overhead. The seeds in sparsified genomics are designed to tolerate more mismatches per read sequence depending on the number of zeros used in the pattern sequence. This might lead to finding a large number of sequences in the reference genome that are similar to the given read sequence hence detecting more genomic variations and possibly some falsely detected variations. We experimentally evaluate and quantify these benefits in detail in the next section.

The Genome-on-Diet framework is a five-step procedure: compressed indexing, pattern alignment, compressed seeding, location voting, and sequence alignment (Fig. 2). These steps can be used individually or collectively depending on the target genomic analysis. The goal of the first step (compressed indexing) is to reduce the size of the genomic sequence and alleviate its overhead. A repeating pattern, called diet pattern, is applied to the reference genome to obtain another version, called patterned genome, that is usually much shorter than the genome sequence (Fig. 2.❶). The seeds collected from the patterned genome are extracted and stored in an index structure, called seed index, along with their start locations in the original (unpatterned) reference genome. The goal of the second step (pattern alignment) is to correctly apply the diet pattern to the query sequence by deciding at which start location the diet pattern can be applied. Multiple right-shifted versions of the same diet pattern are applied to each read sequence to obtain multiple shorter versions, each called patterned read s, of the read sequence (Fig. 2.❷), where $s$ is the shift amount used to shift the diet pattern sequence to the right direction. The seeds collected from each patterned read are extracted and used to query the seed index to calculate the total number of occurrences of all matched seeds per patterned read. An alignment index is calculated based on the shift amount of the corresponding diet pattern that leads to the highest total number of seed occurrences. The goal of the third step (compressed seeding) is to reduce the size of the read sequence and alleviate its overhead. To correctly apply the diet pattern to the read sequence, the diet pattern is first shifted to the right direction by a shift amount equal to the alignment index (Fig. 2.❸). The shifted diet

pattern is then applied to the read sequence, and the resulting patterned read is used to extract seeds and query the seed index to find the potential mapping locations. The goal of the fourth step (location voting) is to find potential mapping locations that lead to high-quality (i.e., highest alignment score) sequence alignment with the read sequence. The list of locations retrieved from querying the seed index is first sorted, and adjacent locations of seed matches are examined to quickly decide whether or not the computationally expensive sequence alignment is needed (Fig. 2.❹). The goal of the last step (sequence alignment) is to calculate the sequence alignment (e.g., the exact number of differences, location of each difference, and their type) between the read sequence and each reference sequence segment at mapping locations that pass the location voting step (Fig. 2.❺).

## Results

We evaluate the performance of Genome-on-Diet for three major widely-used analyses, read mapping, containment search, and taxonomic profiling, using the three prominent sequencing data types (Illumina, HiFi, and ONT reads), different genomes, and large databases.

### Sparsified genomics is a unique mechanism

Our work introduces the concept of sparsifying genomic sequences and processing sparsified sequences in a very fast, efficient, and accurate way. Genome-on-Diet is fundamentally different from other techniques (e.g., spaced seeds[46–51]) that apply patterns to genomic sequences in five important aspects. (1) Genome-on-Diet applies a repeating pattern to the genomic sequence (i.e., reference genome or sequencing read), while spaced seeding keeps the genomic sequence unchanged and applies a pattern to each extracted seed (Fig. 3). (2) The resulting seed in Genome-on-Diet spans a much wider region in the reference genome compared to spaced seeds, which is of vital importance for containment search and metagenomics applications. (3) Genome-on-Diet can use a pattern sequence of any length, while spaced seeding has to use a pattern sequence of length equal to the seed length. (4) Genome-on-Diet avoids extracting seeds that start at a location overlapping with a corresponding 0 in the pattern sequence, while spaced seeding extracts the same number of seeds regardless of the pattern used. For example, using a pattern of '1000101001' results in extracting only 3 overlapping seeds in Genome-on-Diet, while spaced seeding extracts 6 overlapping seeds from the same region (Fig. 3). (5) The resulting seeds in Genome-on-Diet are eventually formed based on different patterns depending on the pattern sequence and the location of the seed with respect to the repeating pattern, while spaced seeding usually applies the same pattern to all

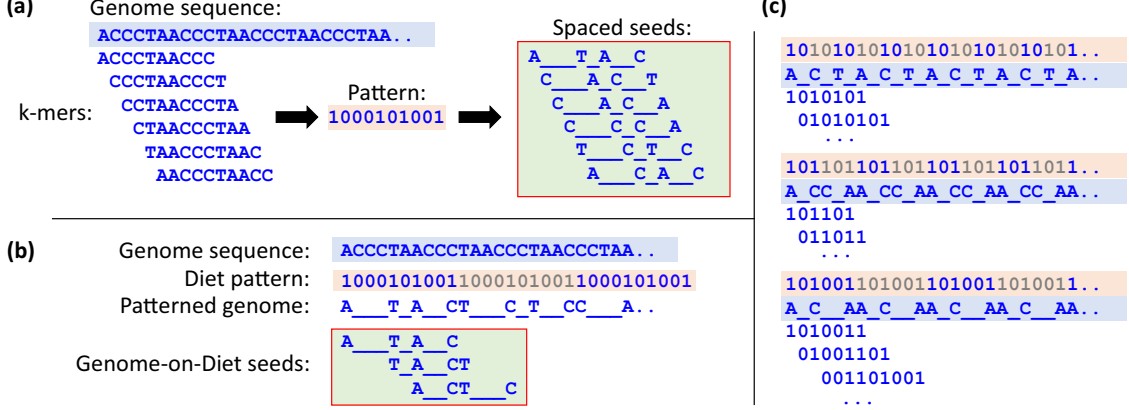

**Fig. 3 | Genome-on-Diet reduces the number of resulting seeds compared to spaced seeds.** Example of resulting seeds after applying (**a**) spaced seeding and (**b**) Genome-on-Diet on a given genome sequence using the same user-defined pattern.

**c** Examples of how each overlapping seed may have its own pattern. The examples at the top, middle, and bottom use user-defined patterns of '10', '101', and '101001', respectively.

seeds. For example, with a pattern of '1000101001', Genome-on-Diet forms the first three seeds with patterns of '1000101001', '0001010011', and '0100110001', respectively, while spaced seeding forms all seeds with a pattern of '1000101001' (Fig. 3). We also provide other examples of how a single user-defined pattern leads to applying different patterns to each seed (Fig. 3c). We experimentally investigate the benefits and downsides of the differences between Genome-on-Diet and spaced seeding.

Unfortunately, we are not aware of any recent read mapper that uses spaced seeding and can compete with the state-of-the-art read mapper, minimap2. To be fair with spaced seeding, we build a computer program that (1) takes a genomic sequence, (2) introduces a copy of the input sequence with a random number of substitutions at random locations such that we evaluate sequence pairs with a wide range of locations and number of differences, (3) extracts genomic seeds from the input sequence and stores them in a list, (4) extracts genomic seeds from the mutated sequence and quantify how many of these seeds appear in the list of stored seeds, and (5) repeats the first four steps for every input genomic sequence. Our computer program uses four different algorithms to extract seeds: all overlapping seeds ("All Seeds" in Fig. 4), minimizer seeds ("Minimizers"), spaced seeds ("Spaced"), and Genome-on-Diet seeds ("Genome-on-Diet"). We use "All Seeds" and "Minimizers" algorithms as a reference for the number of matching seeds. It outputs the Levenshtein distance[68] for each sequence pair along with the seed matching rates calculated when using each of the four seeding algorithms.

We run our computer program using four different seed lengths ($k$), 8, 13, 18, and 21, and a minimizer window of size ($w$) 6, 9, 12, and 11, respectively, using 1000-long sequences. The seed length of 8 is suggested in[69], while the other three seed lengths are within the practical range of seed length used for minimap2[37]. The first three window sizes are calculated as 2/3rds of the seed length based on the reference manual of minimap2 (https://lh3.github.io/minimap2/minimap2.html). The last minimizer window is calculated based on the default preset of minimap2, *-x sr*. We use four pattern sequences ($P$), '110', '10', '101001', and '100' for a seed length of 8 to examine the effect of using different ratios of the number of zeros to the number of ones. For each of the other seed lengths (13, 18, and 21), we use the pattern sequence '10' and another spaced seeding-friendly pattern sequence suggested by the literature. These patterns are '1110110110111'[70] for a seed length of 13, '111001011001010111'[71] for a seed length of 18, and '111101101101011101111'[70] for a seed length of 21. We make our computer program publicly available through the same GitHub project of this work.

We evaluate the four different seeding algorithms, all overlapping seeds, minimizer seeds, spaced seeds, and Genome-on-Diet seeds (Fig. 4). We use two performance metrics: (1) Seed matching rate, which we define as the ratio of the number of matching seeds (i.e., seeds extracted from one sequence that match the seeds extracted from the other sequence) to the total number of extracted seeds. Ideally, the higher the seed matching rate between two sequences the higher the similarity between the two sequences. Thus, a well-performing seeding algorithm should provide an inversely proportional relationship between the seed matching rate and edit distance. (2) Number of accepted sequence pairs, which we define as the number of sequence pairs that have a seed matching rate greater than or equal to a seed matching rate threshold. We determine the seed matching rate threshold as the minimum seed matching rate for all sequence pairs that have an edit distance less than or equal to a specific edit distance threshold. For example, If we want to accept all sequence pairs that have an edit distance threshold of 20, we calculate both the edit distance and the seed matching rate for all input sequence pairs, and we consider the minimum seed matching rate of all sequence pairs

that have an edit distance of at most 20 as the target seed matching rate threshold.

We make four observations. (1) Using a seed length of 8, "Genome-on-Diet" shows higher distinguishability than "Spaced" between sequence pairs with low edit distance and sequence pairs with high-edit distance (Fig. 4a). This is mainly because of the use of multiple different patterns in "Genome-on-Diet" for calculating the extracted seeds (as we exemplify in Fig. 3). Even when tolerating up to two-thirds of the bases of a read (using a pattern '100'), "Genome-on-Diet" still provides distinguishable seed matching rates, while "Spaced" allows most of the compared seeds to be highly similar to each other providing almost no distinguishability (proven theoretically[72]). For example, "All Seeds", "Minimizers", "Spaced", and "Genome-on-Diet" provide that 18%, 24%, 78%, and 19%, respectively, of the input sequence pairs have the same or smaller seed-matching rate to that of a sequence pair with an edit distance of 295. (2) Using a seed length of 8, "Genome-on-Diet" accepts slightly more sequence pairs than both "All Seeds" and "Minimizers" (Fig. 4b). As the number of zeros in the pattern sequence increases, "Spaced" becomes ineffective because it tolerates more differences between compared seeds allowing for very high seed matching rates for both low-edit and high-edit sequence pairs (proven theoretically[72]). For example, "All Seeds", "Minimizers", "Spaced", and "Genome-on-Diet" allow for accepting 34%, 39%, 98%, and 47%, respectively, of input sequence pairs when considering an edit distance threshold of only 138 and a pattern sequence of '100'.

(3) "Genome-on-Diet" still provides high distinguishability between low-edit and high-edit sequence pairs even when using longer seeds and different pattern sequences (Fig. 4c). Using both longer seed length and spaced seeding-friendly pattern improves the distinguishability of "Spaced". (4) For any seed matching rate, "Genome-on-Diet" has a wider range of edit distance values compared to that of "All Seeds", "Minimizers", and "Spaced" (Fig. 4c). This means that "Genome-on-Diet" tolerates more differences than "All Seeds", "Minimizers", and "Spaced". This helps to find more sequence pairs with closely similar edit distance values. This is clearly demonstrated when the number of zeros in the pattern sequence increases (a pattern sequence of '10') (Fig. 4d). In our evaluation, we observe that none of the four seeding algorithms, "All Seeds", "Minimizers", "Spaced", and "Genome-on-Diet", provide false negatives, i.e., none of them reject a sequence pair whose edit distance is less than or equal to the target edit distance threshold.

## Read mapping

Locating possible subsequences of the reference genome sequence that are similar to the read sequence while tolerating differences is still computationally expensive. Tolerating a number of differences is essential for correctly finding possible locations of each read due to sequencing errors and genetic variations. There exists a large body of work trying to tackle this problem quickly and efficiently by using intelligent algorithms, intelligent hardware accelerators, and intelligent hardware/software co-design. Surveys on recent and seminal efforts in these directions can be found in refs. 24,31,32,73.

minimap2[37,74] is the state-of-the-art software read mapper that works well for mapping almost all existing sequencing read types, short, ultra-long, and accurate long reads. minimap2 includes four computational steps, indexing, seeding, chaining, and sequence alignment. First, minimap2 starts with building a large index database using minimizer[38,39] seeds extracted from a reference genome to enable quick and efficient querying of the reference genome. Second, minimap2 uses the same algorithm used for building the index to sketch each read sequence by extracting minimizer seeds. Third, the minimizer seeds extracted from a read sequence are matched to the minimizer seeds extracted from the reference genome. The matching locations are sorted to find adjacent seeds, which are used to build

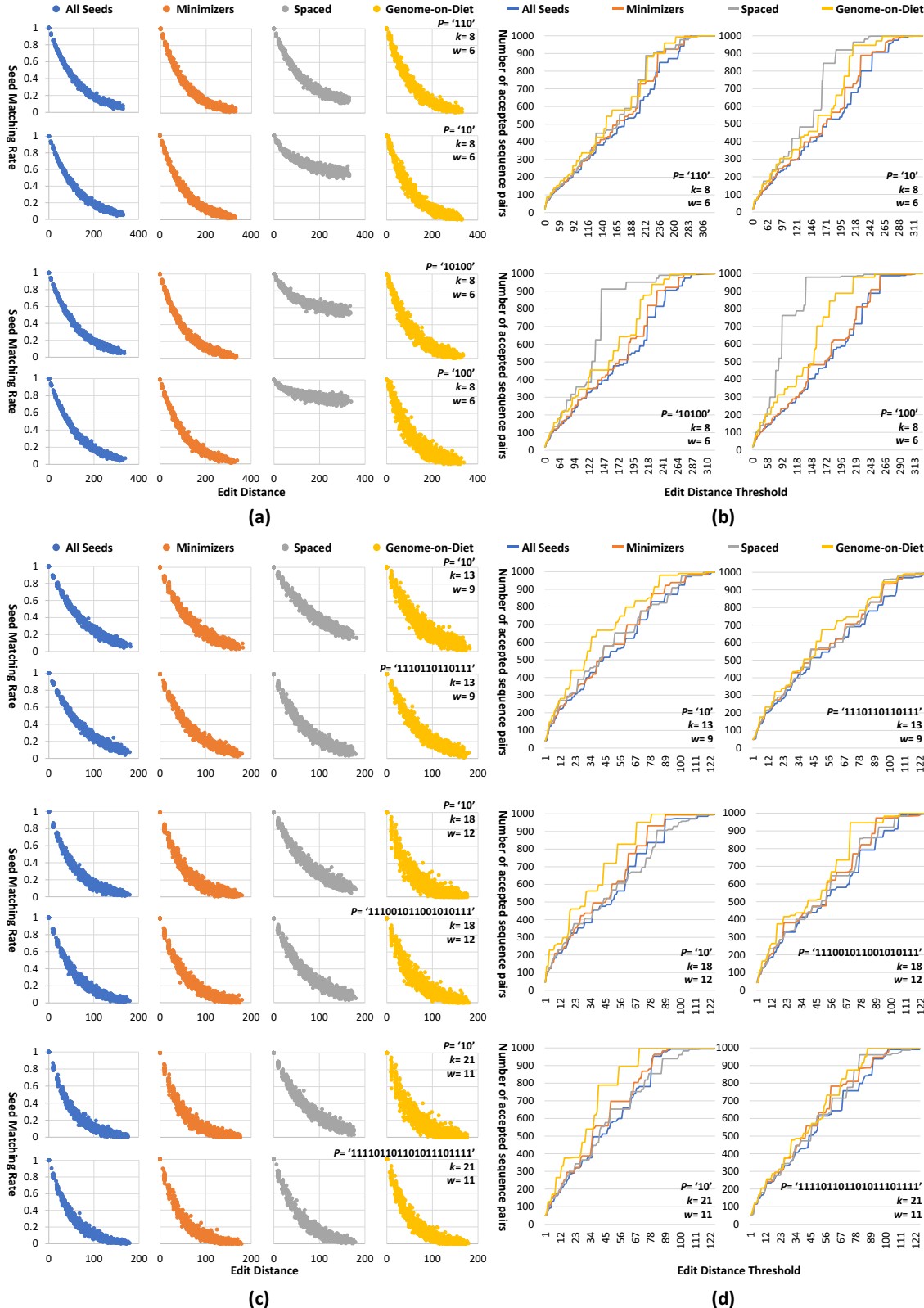

**Fig. 4 | Performance of four seeding algorithms, all overlapping seeds, minimizer seeds, spaced seeds, and Genome-on-Diet seeds.** We evaluate seed matching rates over edit distance when using a seed length of **a** 8 and **c** 13, 18, and 21. We quantify the number of accepted sequence pairs over different edit distance thresholds when using a seed length of **b** 8 and **d** 13, 18, and 21.

chains of matching seeds. Fourth, dynamic programming (DP) based algorithm is used to calculate sequence alignment between every two chains of seeds and generate mapping information into a sequence alignment/map (SAM, and its compressed representation, BAM) file[75].

We observe that read mapping can benefit from sparsified genomics. We investigate using Genome-on-Diet to perform highly optimized and efficient read mapping.

To evaluate the performance of Genome-on-Diet and minimap2, we choose 3 real sequencing read sets for Ashkenazim Son HG002 (NA24385) provided by NIST's Genome in a Bottle (GIAB) project. The three sequencing read sets represent the current state-of-the-art and prominent sequencing technologies, short reads from Illumina, accurate long HiFi reads from PacBio, and ultra-long reads from ONT (Data Availability and Supplementary Table 2). We use the latest version (2.24-r1122) of minimap2 as of 7 June 2022. We use the default presets of minimap2, -x sr, -x map-ont, and -x map-hifi to map short, ultra-long, and accurate long reads, respectively. We use the same parameter values (as provided by minimap2 in `options.c`) for both Genome-on-Diet and minimap2, whenever it is possible and applicable. We vary the window size (-w parameter) for both Genome-on-Diet and minimap2 to evaluate execution time, memory footprint, and the number of mapped reads. We also evaluate the performance of Bowtie2[76] using its latest version (2.5.1). Bowtie2 requires building an index separately before performing read mapping. We run Bowtie2 in two mapping modes: very fast mapping (--very fast) with a small index (default) and very sensitive mapping (--very sensitive) with a large index (--large index). We report the total execution time (system + user time) and memory footprint in all experiments using Linux `/usr/bin/time -v` command. We run all experiments using 40 threads on a 2.3 GHz Intel Xeon Gold 5118 CPU with up to 48 threads and 192 GB RAM.

Demonstrating that a read mapper reports a higher number of mapped reads may not be enough to assess the quality of read mapping as mapped reads may not lead to useful genomic variations. Hence to assess the quality of read mapping results calculated by Genome-on-Diet and minimap2, we use (1) `FreeBayes`[77] and `Sniffles`[78] to call both short insertions/deletions (indels) and structural variations[17], and (2) `Truvari bench`[79] to find common variants between the VCF file of each tool and a recent genomic variation benchmark[80] that reports over 17,000 single-nucleotide variations, 3600 insertions and deletions, and 200 structural variations for human genome reference GRCh38 across HG002. We use the complete Human reference genome GRCh38 (GCA_000001405.15, release date 11 April 2021) to perform read mapping and evaluate genomic variations and complex structural variations that, for example, span two chromosomes.

The first question we need to answer is which pattern sequence a user can choose for sparsified read mapping. To answer this key question, we evaluate the effect of using different pattern sequences on the detected indels and SNPs and the execution time of Genome-on-Diet (Supplementary Table 3). We make five key observations: (1) Compared to non-sparsified read mapping (using Genome-on-Diet with a pattern of '11'), sparsified read mapping can increase the number of correctly detected genomic variations as sparsified read mapping allows tolerating more differences when performing seed matching. (2) The use of pattern '10' provides the best performance when considering all evaluated metrics collectively compared to all evaluated pattern sequences, which is expected based on our analysis ("Sparsified genomics is a unique mechanism"). The use of pattern '10' increases the number of correctly detected variations by 4% and decreases both the number of missed variations and the execution time of read mapping by 25.9% and 28.4%, respectively, compared to that provided by the non-sparsified read mapping. (3) Regardless of the pattern used, sparsified read mapping has the drawback of increasing the number of incorrectly detected variations. This is because of the ability of Genome-on-Diet to tolerate more differences between seeds compared to other methods (e.g., minimizer seeds) that use non-sparsified (contiguous) seeds. This leads to detecting more mapping locations per read and, hence, more variations (both true and false) to be detected. This can be possibly addressed by applying quality filtering mechanisms that, for example, examine the number of matching bases within the matching seeds instead of only quantifying the matching seeds (as in our location voting step) since

the seeds are sparsified. (4) The location of zeros in the pattern sequence has a slight effect on all evaluated metrics. That is, the patterns '110', '101', and '011' (similarly patterns '101001', '100101', '001101') all provide similar performance. However, the number of zeros in the pattern sequence has a significant effect on all evaluated metrics. (5) Though both '10' and '101001' patterns lead to excluding half of the bases from the reference genome and read sequences, they lead to different read mapping performances. This is mainly because each pattern may result in applying different patterns to each seed depending on the location of the extracted seed in the diet pattern. For example, the pattern '10' results in applying one of the two patterns, '101010...' or '01010...', to each overlapping seed, while the pattern '101001' results in applying one of the three patterns, '1010011...', '01001101...', or '001101001...', to each overlapping seed (Fig. 3c). This directly affects the number of potential mapping locations that need to be examined, which in turn can affect the mapping results.

We conclude that the use of pattern '10' still provides the best accuracy-speedup tradeoffs for sparsified short read mapping. We also make the same observation for long read mapping (Supplementary Table 4). Thus, we decide to use the pattern '10' for the read mapping application.

We evaluate the execution time and memory footprint of Genome-on-Diet and minimap2 for performing read mapping (Fig. 5). We use three read sets provided by the three prominent sequencing technologies, Illumina, HiFi, and ONT. We use the pattern of '10' based on our previous discussion. We make six key observations. (1) Genome-on-Diet is always faster than minimap2 (Fig. 5A–C). Genome-on-Diet is 2.57–5.38×, 1.13–2.78×, and 3.52–6.28× faster than minimap2 using Illumina, HiFi, and ONT reads, respectively. We provide exact values in Supplementary Table 5. (2) Sparsifying genomic sequences along with the proposed optimization strategies accelerate every step of read mapping (Fig. 5D–F). Building the index for the complete human reference genome using Genome-on-Diet is 1.79–1.85× faster than that using minimap2. Generating minimizer seeds (using both pattern alignment and compressed seeding steps) from sequencing reads using Genome-on-Diet is 1.72–2.69× faster than that using minimap2. Detecting potential mapping locations using Genome-on-Diet is 65.57–651.19× faster than that using minimap2 as Genome-on-Diet does not use computationally expensive chaining step. The sequence alignment becomes the most computational step in Genome-on-Diet as it accounts for 55%, 91%, and 81% (compared to 14%, 58%, and 17% in minimap2) of the total execution time using Illumina, HiFi, and ONT reads, respectively. This means that Genome-on-Diet can benefit (based on Amdahl's law) from exploiting existing and future software/ hardware acceleration approaches[31,32] of sequence alignment, such as Darwin[81], GenASM[30], SeGraM[82], and WFA-GPU[83]. (3) Genome-on-Diet shows 1.6–1.73× and 2–2.1× less memory footprint than minimap2 using Illumina and HiFi reads, respectively, (Fig. 5G, H). Genome-on-Diet shows 1.17–1.46× higher peak memory footprint than that of minimap2 using ONT reads (Fig. 5I), which is mainly due to our design choice of performing sequence alignment for the complete genomic sequence instead of only performing sequence alignment between every two chains as in minimap2. We observe that the maximum read length of the ONT reads we are using is 1,331,423 bases, which makes the peak memory footprint of Genome-on-Diet excessively high (>190 GB) when processing ultra-long ONT reads. To allow for only a reasonable peak memory footprint, we split the ultra-long reads into segments of a fixed length, each of which is at most 30,000 base long. Alternatively, such a shortcoming can be addressed using recent sequence aligners that have low-memory footprint (e.g., BiWFA[84]). (4) The performance of Genome-on-Diet is nearly consistent over different values of window size ($w$), due to two reasons: the independence of the location voting step on both the length of the seeds and the distance between every two seeds as in minimap2, and the parallel execution of compressed indexing, pattern alignment, and

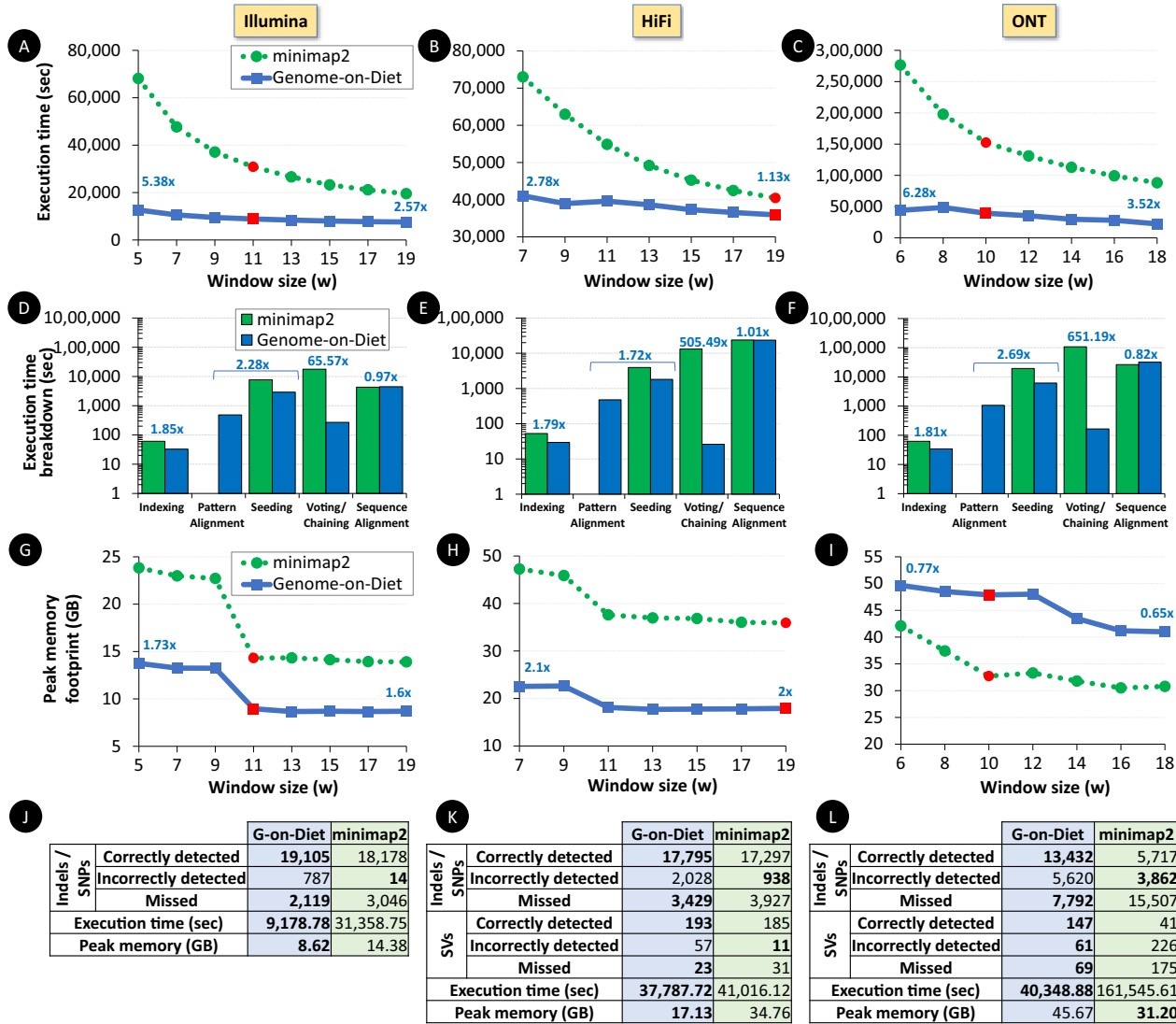

**Fig. 5 | Read mapping performance and quality (genomic variation detection) of Genome-on-Diet and minimap2 using three real read sets, Illumina, HiFi, and ONT.** We use minimap2's default $k$ value for both tools, which is 21, 19, and 15, for Illumina, HiFi, and ONT reads, respectively. The red data point in each plot represents minimap2's default $w$ value. Execution time for performing read mapping using (A) Illumina, (B) HiFi, and (C) ONT reads. Execution time of each step of Genome-on-Diet and minimap2 when using minimap2's default k and w values for both tools for mapping (D) Illumina, (E) HiFi, and (F) ONT reads. The steps are indexing, pattern alignment (only for Genome-on-Diet), seeding, voting/chaining, and sequence alignment. The values (in blue) represent the speedup provided by each step of Genome-on-Diet over the corresponding step of minimap2. Peak memory footprint of Genome-on-Diet and minimap2 for performing read mapping for (G) Illumina, (H) HiFi, and (I) ONT reads. Number of indels/SNPs and SVs as provided by Genome-on-Diet and miniamp2 when using the minimap2's default k and w values for (J) Illumina, (K) HiFi, and (L) ONT presets. We highlight the best result in bold per each evaluation metric.

compressed seeding. This makes Genome-on-Diet more suitable for choosing different window sizes without adding additional overhead. (5) Genome-on-Diet shows better performance when using a window size value that is different from the default value of minimap2. For example, users can use a window size of 5, 7, and 6 for Illumina, HiFi, and ONT reads, respectively, to obtain the lowest execution time. (6) Genome-on-Diet is 5.24× and 17.26× faster than Bowtie2[76] in "very fast" mapping mode and "very sensitive" mapping mode, respectively, (Supplementary Table 6). Bowtie2 provides 1.56–2.1× lower peak memory footprint (as Bowtie2 uses FM-index, which results in a compressed, small index based on the Burrows–Wheeler transformation) and 4.2–5.7× higher storage footprint than Genome-on-Diet.

We conclude that Genome-on-Diet is memory-frugal and it improves the execution time of read mapping by 1.13–6.28× using reads from the three prominent sequencing technologies, Illumina, HiFi, and ONT.

To evaluate the quality of the read mapping results provided by Genome-on-Diet and minimap2, we investigate the quality of the mapped reads by examining the number of detected genomic variants, single-nucleotide polymorphisms (SNPs), insertions/deletions (indels), and structural variations (SVs), that can be detected from the output alignments of Genome-on-Diet and miniamp2 (Fig. 5J–L). We make three observations. (1) Sparsifying genomic sequences (i.e., using only half the number of bases) does not lead to any loss in the end results of read mapping. Using sparsified seeds always leads to detecting a higher number of SNPs, indels, and SVs compared to minimap2 as we expect, while Genome-on-Diet is also faster than minimap2. (2) Genome-on-Diet provides both the highest number of true genomic variations (of all types) and the least number of missed variations (of all types) compared to minimap2 based on the ground truth variation benchmark[80]. This is also true for Genome-on-Diet over Bowtie2

(Supplementary Table 6). (3) Similar to what we observe when evaluating different pattern sequences ("Read mapping"), Genome-on-Diet still comes with the drawback of increasing the number of incorrectly detected variations of all types.

We conclude that Genome-on-diet is very fast, memory-frugal, and its read mapping results lead to the detection of a higher number of true genomic variations and a lower number of missed variations compared to the state-of-the-art read mapper, minimap2.

## Containment search

Containment search is typically used to measure the similarity between two genomic data sets by calculating k-mer intersection between their respective k-mer sets[57,85,86]. The theoretical concept of containment search is useful for several key genomics and metagenomics applications, such as identifying a small number of candidate organisms that are potentially present in the metagenomic sample[21], identifying de novo variations[62], and directly comparing reads sequenced from normal and tumor genomes to identify somatic variations[18].

Containment search is typically two steps, building a containment index for one dataset and finding k-mer intersection between the other dataset and the index. CMash[57] is one of the state-of-the-art hashing-based approaches for building the containment index between two genomic sets. CMash uses a k-mer ternary search tree (KTST) that supports storing and querying k-mers of variable sizes without the need for reconstructing different KTST for each k-mer size, which is claimed to provide high sensitivity and accuracy[57]. CMash[57] is used in Metalign[21] along with a state-of-the-art k-mer counting technique, called KMC3[56], to quickly identify a subset of candidate organisms in large microbial reference databases (e.g., RefSeq) that share similarity with a given sequence read set. We observe that CMash and KMC3 together generate more than 7× the size of the examined reference database (e.g., the size of RefSeq database is >2.7 Tbp doubles nearly every 3 years) as auxiliary data used for the similarity measurement. Although the vast majority of such generated data is built only once, it requires significant storage capacity and long memory access time to accommodate and use the generated data. We investigate using Genome-on-Diet to replace both KMC3 and CMash (referred to as KMC3+CMash) to find the similarity between sparsified sequences as Genome-on-Diet is designed to demand only a small memory footprint and without the need for pre-built indexes.

We perform two key experimental evaluations, performing large-scale containment search and replacing KMC3+CMash. For the first experimental evaluation, we prepare four different reference databases. (1) The complete Human Genome GRCh38.p14 (GCF_000001405.40, release date 3 February 2022) with a file size of 3.4 GB. (2) The largest sequenced reference genome, Pinus Taeda[87] (also known as loblolly pine, GCA_000404065.3, release date 9 January 2017) with a file size of 28.4 GB. (3) We empirically choose 1809 FNA files, with 13,768,320 strains/contigs, from RefSeq with a total size of 97.2 GB. We call this reference database RefSeq1. (4) We duplicate RefSeq1 database to obtain a larger reference database with a file size of 194.4 GB. We call this reference database RefSeq2. We simulate 100,000 HiFi-like reads using `wgsim` (v1.0 conda) from each of the four databases. The well-known HiFi read simulator, PBSIM, does not work using a very large number (more than about 9000 contigs) of input sequences. We configure Genome-on-Diet to use `k` = 19 and `w` = 16 with five different pattern sequences, '11' (representing no sparsifying), '1110', '110', '10', and '100'. To keep the peak memory footprint of Genome-on-Diet below the maximum main memory (192 GB RAM), we limit the loaded number of bases at once to 30 billion bases, using the `-I` parameter.

For the second experimental evaluation, we compress each reference genome in RefSeq1 using the `gzip` tool since Metalign accepts only compressed reference files, which reduces the total size

of the FNA files down to 30.2 GB. We use two metagenomic read sets from the widely cited comprehensive benchmarking study, the Critical Assessment of Metagenome Interpretation (CAMI)[22,88]. The two read sets are RL_S001_insert_270.fq from CAMI Low, which has 99,796,358 reads (14,969,453,700 bases), and RH_S001_insert_270.fq from CAMI High, which has 99,811,870 reads (14,971,780,500 bases). We use Metalign's conda package (version 0.12.5) to run CMash[57] and KMC3[56]. We use the script provided by Metalign in[89] to retrain and build the database of CMash and dump all k-mers calculated by KMC3[56] for the reference genome. For a fair comparison with Genome-on-Diet, we slightly changed the script so that `MakeStreamingDNADatabase.py` uses multithreading using the parameter -t. We disable `map_and_profile` step in Metalign to enable calculating only the containment search step between the CAMI data and the chosen reference database. We activate the precise mode of Metalign, which allows Metalign to provide low falsely accepted species. We also run the complete Metalign to obtain the list of actual organisms that are present in each metagenomic sample for obtaining the ground truth results for evaluating the accuracy of KMC3+CMash and Genome-on-Diet. We disable sequence alignment from Genome-on-Diet as sequence alignment is not relevant for containment index calculation. We define mapped read in this context as the read that receives a mapping location using the location voting step. We empirically configure Genome-on-Diet to output the species genome sequence (in FNA format) that has at least 100,000 mapped reads with a genome coverage of at least 1. We use two configurations for Genome-on-Diet to examine the tradeoffs between increasing the memory footprint and reducing the execution time. We refer to these configurations as 10 Gb and 20 Gb, where we allow Genome-on-Diet to load batches of data, each of which has at most 10 billion and 20 billion bases into the main memory, respectively. We empirically use the pattern of '10', $k$ = 28, and $w$ = 40. We run all experiments using 40 threads on a 2.3 GHz Intel Xeon Gold 5118 CPU with up to 48 threads and 192 GB RAM. We report the total execution time (system + user time) and memory footprint in all experiments using Linux `/usr/bin/time -v` command.

We analyze the benefits of sparsifying genomic sequences using Genome-on-Diet for performing containment searches without the need to pre-build huge indexes in advance. Different pattern sequences can be used to sparsify the genomic sequences, which affects the number of included and excluded bases from input sequences (Fig. 6). We make two key observations. (1) Sparsifying genomic sequences makes large-scale containment search feasible and efficient. The use of '1110', '110', '10', and '100' patterns accelerates containment search (both containment indexing and k-mer intersection) by 1.3×, 1.4×, 1.9×, and 2.7×, respectively, showing a reduction in the execution time by almost ¼, 1/3, ½, and 2/3, respectively, (Fig. 6A). This demonstrates that the execution time scales linearly with the number of zeros determined in the pattern sequence. The use of '1110', '110', '10', and '100' patterns directly reduces the size of the index by ¼, 1/3, ½, and 2/3, respectively (Fig. 6B). The peak memory footprint is also reduced (Fig. 6D), but at fewer rates than that with the index size as the peak memory is affected by the used pattern sequence besides other factors such as the size of the reference genome, occurrence frequency of each seed, and the amount of data loaded at once. (2) Sparsifying genomic sequences reduces the maximum number of votes (i.e., seed matches that vote on the same mapping location) per read. Reducing the size of the index by only ¼ reduces the number of votes by, on average, 3.68× for a small reference database and by, on average, 1.9× for a large reference database (Fig. 6C). This is expected as Genome-on-Diet provides low minimizer density. The minimizer density is the expected number of extracted minimizers divided by the length of the genomic sequence[90]. Lower density is desirable as storing fewer minimizers in the hash table implies a reduction in the execution time and memory footprint of querying the index and performing computation on the retrieved locations.

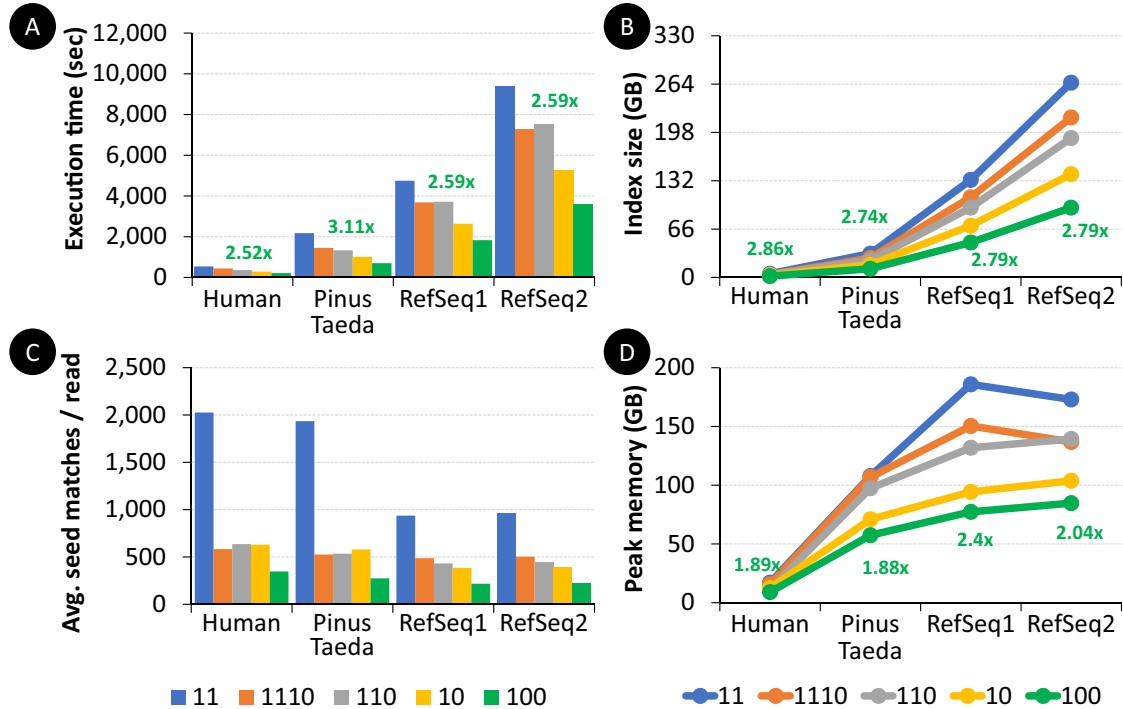

**Fig. 6 | Effect of sparsifying genomic sequences on large-scale containment search.** We use five pattern sequences. The '11' pattern represents no sparsifying is performed, and the other four patterns, '1110', '110', '10', '100', represent different degrees of sparsifying. **A** Execution time of performing both containment indexing and k-mer intersection using human genome, Pinus Taeda, RefSeq1, and RefSeq2 without the need for pre-building their indexes. **B** The size of the containment index. **C** The average number of seed matches per each input read sequence using different large genomic data and different patterns. **D** The peak memory footprint when performing both containment indexing and k-mer intersection. The values (in green) represent the improvement that is gained from sparsifying the genomic sequences using a pattern sequence of '100' compared to non-sparsified (using a pattern of '11') genomic sequences.

We conclude that Genome-on-Diet is very beneficial for both small-scale and large-scale containment search as it allows sparsifying genomic sequences differently for different applications using user-configurable pattern sequences.

Based on the benefits of sparsifying genomic sequences using Genome-on-Diet that we demonstrate, we explore using Genome-on-Diet to replace the state-of-the-art tool, KMC3+CMash, for performing containment search (Table 1). We make four key observations. (1) Genome-on-Diet, using $k = 28$ and $w = 40$, is 72.7–75.88× (45.3–48.2× using a data batch size of 10 billion bases) faster and 723.3× more storage-efficient than the end-to-end KMC3+CMash for performing both containment indexing and k-mer intersection, using a data batch size of 20 billion bases. When the indexing step is not considered in the total execution time, then Genome-on-Diet becomes 1.62–1.9× (1.1–0.14× using a data batch size of 10 billion bases) faster than KMC3+CMash using a data batch size of 20 billion bases. (2) Genome-on-Diet has no storage usage (0 GB) as it does not provide any additional data. KMC3+CMash provides 711.3 GB of data during indexing and about 12 GB of data during k-mer intersection step. Genome-on-Diet does not require to pre-build the index, dump it to the storage, and load it again for k-mer intersection, which is one of the key reasons behind the saving in both the total execution time and storage space usage. (3) It is still possible to pre-build the index using Genome-on-Diet, which takes about 3902 and 1590 seconds to build and results in providing a 153.5 GB and 78.9 GB of index with a memory footprint of 20.7 GB and 7.5 GB, using two configurations ($k = 21$, $w = 11$) and ($k = 28$, $w = 40$), respectively, using a data batch size of 10 billion bases. The execution time of the indexing step of KMC3+CMash is 38.8–43.7× higher than the execution time of their k-mer intersection step. (4) We observe a similar indexing performance to that of KMC3+CMash in even the well-maintained and widely-used Kraken2[28] (Supplementary Table 7).

We conclude that sparsifying genomic sequences saves the end-to-end execution time and storage space of containment search. Though the benefits provided by Genome-on-Diet scale linearly with the number of zeros determined in the pattern, Genome-on-Diet provides the additional key benefit of building the index on-the-fly without requiring a large peak memory footprint. The state-of-the-art tools, KMC3+CMash and Kraken2, that perform containment search produce an index of RefSeq genomes (3.5 TB) equal to 24.5 TB (7 × 3.5 TB) and 7 TB (2 × 3.5 TB), respectively. While building the index using these state-of-the-art tools takes much longer than that of Genome-on-Diet as we demonstrate, loading and querying such a huge index from storage into the main memory is another concern for the state-of-the-art tools. Challenges with TB-scale databases need to be evaluated in future work.

We evaluate the accuracy of Genome-on-Diet and KMC3+CMash in measuring the similarity between two genomic data sets (Supplementary Table 8). We define the accuracy in this context as both the number of truly accepted strains/contigs and the number of falsely accepted strains/contigs compared to the ground truth results of Metalign, a state-of-the-art taxonomy profiling tool. We make sure that the truly accepted strains/contigs are exactly the same as those reported by Metalign using their reported taxonomic identifiers (taxid). We make three key observations. (1) Both Genome-on-Diet and KMC3+CMash do not miss any similar data (i.e., a true accept rate of 100%). (2) Genome-on-Diet shows a false accept rate of less than 0.0005%, which is acceptable for two main reasons: (1) It is significantly low and (2) Tools use containment index usually use another step for providing base-level alignment with 0% false accept rate. (3) There also exists a large number of sketching algorithms that are typically used for estimating the similarity distance between two genomic sequences such as Dashing[91] and BinDash[61]. We evaluate the performance of using BinDash[61] for containment search as BinDash also allows for quickly measuring the

**Table 1 | Performance, memory footprint, and storage usage of KMC3+CMash and Genome-on-Diet for containment indexing and k-mer intersection**

|  |  | Indexing time (sec) (User+Sys) | Indexing memory (GB) | Indexing storage (GB) | k-mer intersection time (sec) | k-mer intersection memory (GB) | k-mer intersection storage (GB) |
|---|---|---|---|---|---|---|---|
| KMC3+CMash | CAMI Low | 456,078 | 11.7 | 711.3* | 11,752 | 12.95 | 12 |
|  | CAMI High | 456,078 | 11.7 | 711.3* | 10,418 | 11.7 | 9.2 |
| Genome-on-Diet (10 Gb) | CAMI Low | 0 | 0 | 0 | 10,318 | 18.59 | 0 |
|  | CAMI High | 0 | 0 | 0 | 9687 | 18.66 | 0 |
| Genome-on-Diet (20 Gb) | CAMI Low | 0 | 0 | 0 | 6165 | 35.17 | 0 |
|  | CAMI High | 0 | 0 | 0 | 6415 | 36.59 | 0 |

*The index of KMC3 + CMash includes:kmc_suf,.kmc_pre,.h5,.fa,.bf,.desc,.tst files.

similarity without the need for a pre-built index. We observe that Bin-Dash shows a similar k-mer intersection time to that of Genome-on-Diet using a data batch size of 10 billion bases (Supplementary Table 9), however, BinDash provides a large false accept rate and a considerable number of falsely rejected strains (Supplementary Table 10).

We conclude that Genome-on-Diet provides a very low false accept rate, the lowest end-to-end execution time, and the lowest storage space usage. Hence, it is very effective for measuring the similarity between two genomic data sets.

### Taxonomic profiling

Identifying the presence and relative abundances of microbes in an environmental sample (recovered directly from its host environment) efficiently and accurately remains a daunting challenge[1,21,92–94]. Such an identification can be computationally performed using a taxonomic profiling step, which is a critical first step in microbiome analysis. Existing analysis techniques require comparing the genomic composition of the subject sample to a large volume of genomic data and using computationally expensive algorithms for identifying a wide range of microbes. This necessitates performing the analysis on only high-performance computing platforms that are normally power-hungry and nonexistent in remote areas, small facilities, and outer space[95,96]. We believe that there is still a huge need and space for improving existing metagenomic analysis tools.

Metalign is the state-of-the-art mapping-based metagenomic analysis tool[21]. Metalign employs three key steps. Metalign uses as the first step KMC3 and CMash together to narrow down the list of candidate organisms that are potentially present in the metagenomic sample. Metalign uses minimap2 as the second step to accurately map metagenomic reads to the filtered candidate genomes. Finally, Metalign estimates the relative abundances of microbes in the sample by combining information from reads that uniquely map to one genome with those that align to multiple genomes. We investigate using Genome-on-Diet to improve Metalign as Genome-on-Diet is highly optimized for performing accurate and memory-frugal containment indexing and read mapping without the need for pre-building indexes.

We build on top of our containment search evaluation ("Containment search") by using KMC3+CMash or Genome-on-Diet as the first step for narrowing down the list of 1809 species, with 13,768,503 strains/contigs, to only these that share similarity with the input metagenomic reads. The strains that KMC3+CMash or Genome-on-Diet accepts are used for performing the second and third steps of Metalign. We use two metagenomic read sets, RL_S001_insert_270.fq and RH_S001_insert_270.fq, from CAMI[22,88]. We use Metalign's conda package (version 0.12.5). We disable select_db (which is performing KMC3+CMash) in metalign.py so that only taxonomic profiling is executed. We run Metalign to obtain the list of actual organisms that are present in each metagenomic sample, along with their relative abundance, for evaluating the accuracy of taxonomic profiling. We measure the correctness using the F1 score[97] and the accuracy using the L1 norm error[97]. The F1 score[97] is the harmonic average of precision (number of truly reported taxa over the number of all reported taxa) and recall (number of truly reported taxa over the sum of the number of truly reported taxa and number of falsely reported taxa). The L1 norm error measures the calculation accuracy of the relative abundance of taxa in a sample, and it is calculated as the sum of the absolute differences between the true and predicted relative abundances[97].

We disable the recovery mode in Genome-on-Diet for taxonomic profiling as the recovery mode forces each read sequence to map to one of the reference genomes stored in the reference database. This affects the profiling results and especially the relative abundance calculation as it depends on the number of mapped reads to each taxa. Thus, Genome-on-Diet for taxonomic profiling considers only reads with a sufficient number of matching seeds for further analyses. We configure both Genome-on-Diet and minimap2 of Metalign to use a

**Table 2 | End-to-end performance, memory footprint, and storage space usage of Metalign and Genome-on-Diet for metagenomic analysis**

| Containment indexing algorithm | Taxonomic profiling algorithm | Indexing time (sec) | k-mer intersection time (sec) | Taxonomic profiling time (sec) | Total (sec) | Memory footprint (GB) | Storage usage (GB) |
|---|---|---|---|---|---|---|---|
| CAMI low | | | | | | | |
| Metalign | | 456,078 | 11,752 | 3114 | 470,885 | 12.95 | 723.3 |
| KMC3+CMash | Genome-on-Diet | 456,078 | 11,752 | 2473 | 470,303 | 12.95 | 723.3 |
| Genome-on-Diet | Genome-on-Diet | 0 | 6165 | 2532 | 8697 | 35.17 | 0 |
| Genome-on-Diet | Metalign | 0 | 6165 | 3245 | 9410 | 35.17 | 0 |
| CAMI high | | | | | | | |
| Metalign | | 456,078 | 10,418 | 1504 | 468,000 | 11.74 | 720.5 |
| KMC3+CMash | Genome-on-Diet | 456,078 | 10,418 | 1220 | 467,716 | 11.74 | 720.5 |
| Genome-on-Diet | Genome-on-Diet | 0 | 6414 | 1149 | 7563 | 36.59 | 0 |
| Genome-on-Diet | Metalign | 0 | 6414 | 1503 | 7917 | 36.59 | 0 |

k-mer length of 28 and a minimizer window length of 40. We run all experiments using 40 threads on a 2.3 GHz Intel Xeon Gold 5118 CPU with up to 48 threads and 192 GB RAM. We report the total execution time (system + user time) and memory footprint in all experiments using Linux `/usr/bin/time -v` command.

We analyze the benefits of using Genome-on-Diet as taxonomic profiler for metagenomic samples (Table 2), as Genome-on-Diet is carefully designed to build large indexes on-the-fly. We make a key observation. We make two key observations. (1) Sparsifying genomic sequences accelerates the end-to-end taxonomic profiling by 54.15–61.88× and reduces its storage usage by at least 720×. When the indexing step is not considered in the total execution time, Genome-on-Diet becomes 1.58–1.71× faster than Metalign. (2) The memory footprint of Genome-on-Diet is 2.7–3.12× higher than that of Metalign as Genome-on-Diet performs containment indexing using large data batches of size 30 billion bases each. The batch size can be reduced such that Genome-on-Diet provides a comparable memory footprint to that of Metalign (Table 1) at the cost of a slight increase in execution time. For example, using a batch size of 10 billion bases reduces the peak memory footprint from 35.17 GB to only 18.59 GB while increasing the total execution time of taxonomic profiling from 8697 seconds to 12,850 seconds (using CAMI Low), which is still 36.6× (470,885/12,850) faster than that of Metalign.

We conclude that Genome-on-Diet provides benefits in terms of execution time and storage space usage for taxonomic profiling and end-to-end metagenomic analyses.

In all experiments performed for taxonomic profiling, we verify the correctness and accuracy of each algorithm (or combination of algorithms) by comparing the presence/absence of each taxon reported by the subject algorithm to that of Metalign. We always observe the same presence/absence of each taxon in all taxonomic profiles calculated by each algorithm. Hence, the F1 score[97] for all combinations of algorithms always equals 1. However, we observe slight differences between the relative abundance estimates of Metalign (assuming the taxonomy profile of Metalign is the ground truth) and other algorithms due to the different numbers of mapped reads provided by Genome-on-Diet and minimap2 during the metagenomic read mapping step. Such a difference does not affect the order of each taxon in the taxonomic profile for any of the evaluated combinations of algorithms. We measure these differences and represent them as L1 norm error (Supplementary Table 11), which measures the accuracy of reconstructing the relative abundance of taxa in a sample[97]. We observe that the L1 norm error is very negligible for the evaluated combinations of containment index and taxonomic profiling algorithms. The differences are mainly due to the ability of Genome-on-Diet to tolerate more differences when matching seeds. This leads to obtaining more mapping locations (in one or more reference genomes) per read

("Sparsified genomics is a unique mechanism"), which directly affects the relative abundance calculations.

We conclude that Genome-on-Diet correctly and accurately identifies the presence, absence, and relative abundance of taxa in a metagenomic sample.

## Discussion

Searching reference genomes and databases for sequences similar to sequencing reads is still extremely challenging due to the large size of analyzed data and technological limitations of modern sequencing platforms. We introduce the concept of sparsified genomics where genomic applications and analyses use the minimum required number of bases from genomic sequences to find their shared similar regions and exhaustively search large databases. We demonstrate the benefits of sparsified genomics using a highly efficient, highly optimized framework, called Genome-on-Diet. Genome-on-Diet requires smaller computational resources such that it can enable performing genomic analyses on resource-limited mobile devices in remote areas, small facilities, and outer space. We show that sparsified genomics may falsely report dissimilar sequences as similar ones. This is because the sparsified genomics concept allows for tolerating differences between seeds depending on the number of zeros used in the pattern sequence. We experimentally evaluate such benefits and drawbacks in this article. The optimization strategies and the concept of sparsified genomics can be exploited and leveraged to improve existing algorithms and applications. Our work has broad applicability as we demonstrate benefits in read mapping, containment search, and robust microbiome discovery. In addition, other potential applications are pangenome mapping[3,98], quantification of transcript expression[99], identifying de novo variations by directly comparing read sequences between related individuals[62], identifying somatic variations by directly comparing reads sequenced from normal and tumor genomes[18], and pre-alignment filtering[100]. We hope that these efforts and the challenges we discuss provide a foundation for future work in catalyzing existing genomic analyses and enabling new analyses. We anticipate that our work will be a valuable resource for many academic and industrial research groups performing small- and large-scale genomic analyses.

## Methods

The primary purpose of Genome-on-Diet is to reduce the end-to-end execution time and memory footprint of the indexing and seeding steps in genomic analyses by sparsifying genomic sequences and enabling processing of the sparsified, shorter genomic sequences. Given two genomic sequences, a reference sequence $R[0, ..., r-1]$ and a query sequence $Q[0, ..., q-1]$, where $r \geq q$. The two genomic sequences are sequences of A, C, G, T in the DNA alphabet ($\Sigma = \{A, C, G, T\}$) in addition to the ambiguous base, N. Suppose the user's pattern sequence is $P[0, ..., p-1]$ consisting of only ones and zeros ($\Sigma = \{0, 1\}$),

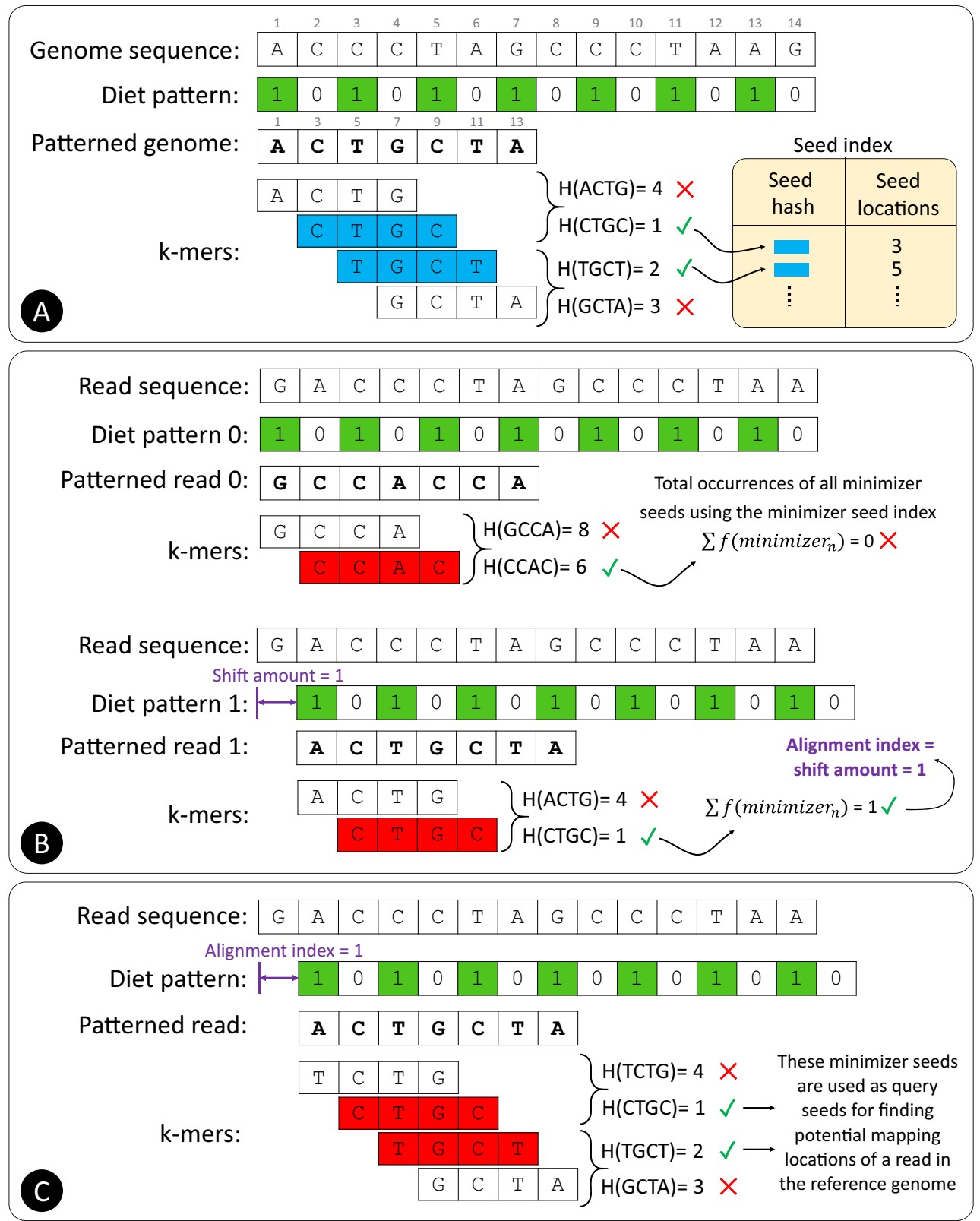

**Fig. 7 | A walkthrough of the first three Genome-on-Diet steps.** **A** Compressed indexing. We assume a user's pattern of '10', which is repeated 7 times to build the diet pattern and accommodate the input genome sequence. As a result of applying the diet pattern to the genome sequence, the patterned genome sequence is half in length compared to the input genome sequence, $\beta = 2$. The seed length, $k$, is 4 and minimizer window, $w$, is 2. H() is a hash function that takes a *k-mer*, subsequence of length $k$, and provides a hash value. **B** Pattern alignment. We generate two diet patterns and two patterned read sequences. From each patterned read sequence, we only collect a single minimizer. By examining the total occurrence frequency of each minimizer, we find out that the alignment index, $a$, equals the shift amount used for patterned read 1 (i.e., $a = 1$). **C** Compressed seeding. We align the pattern with the read sequence and calculate the minimizer seeds of all possible overlapping k-mers to query the seed index to obtain matching locations.

where $r \geq q \gg p$. The number of ones in the user's pattern is called weight, $x$. The reduction ratio of a genomic sequence is $\beta = p/x$. Our goal is to quickly, memory-efficiently, and accurately find all correct mapping locations of $Q$ in $R$ by matching a set of seeds from the patterned genome, $PR[0, …, pr - 1]$, with these extracted from the pattern read, $PQ[0, …, pq - 1]$ and performing base-level alignment, where $r \geq pr \geq q \geq pq \gg p \geq x$. $PR$ is a subset that includes all non-zeros elements of the result set of multiplying the repeating user's pattern with the reference sequence, $PR \subseteq \{R[i].P[i\bmod p]\}_{i=0}^{r-1}$. $PR$ is a subset that includes all non-zeros elements of the result set of multiplying the repeating user's pattern with the query sequence, $PQ \subseteq \{Q[i+a].P[i\bmod p]\}_{i=0}^{q-1}$, where $a$ is the alignment index. We next explain how to achieve our goal using the five main steps, compressed indexing, pattern alignment, compressed seeding, location voting, and sequence alignment, of the Genome-on-Diet algorithm (Figs. 2 and 7).

## Compressed indexing

The compressed indexing step takes a reference genome as an input and provides a seed index as output (Fig. 7A). The input reference genome sequence is preserved throughout the entire workflow of the Genome-on-Diet algorithm, which is important for performing a base-level alignment. Genome-on-Diet alters only a copy of the input reference genome for building the seed index. The seed index is built in three steps. (1) We use a user-defined binary pattern to identify the location and number of the to-be-dropped bases. The user provides a binary pattern sequence, $P$, of any length, $p$, and weight, $x$. Genome-on-Diet considers the user-defined pattern as the shortest repeating substring and uses a repeating version of the pattern sequence to build the diet pattern of length equals the length of the subject genomic sequence. (2) Genome-on-Diet examines the diet pattern, and whenever it encounters a 1 in the diet pattern, it stores the corresponding base of the reference genome in the patterned genome sequence. The length of the patterned genome sequence equals the total number of 1's in the diet pattern. (3) Genome-on-Diet computes minimizers of the patterned genome sequence and indexes them in a hash table, with the key being the hash value of a minimizer seed and the value is a list of start locations of all minimizers whose hash values are the same. The location stored in the hash table is the start location of each minimizer in the original unpatterned reference genome. This facilitates direct identification of the mapping location for sequence alignment without the need for an additional location translation (from patterned genome to reference genome) step.

Genome-on-Diet follows the same approach of minimap2[37] to 1) store a sorted list of minimizer hashes and their sorted locations, 2) calculate the double-strand $(w, k)$-minimizer of a string is the smallest k-mer in a surrounding window of $w$ consecutive k-mers[38,39]. We do not directly insert minimizer information (hash value and location) into the hash table. Instead, we append the minimizer information to an array and sort the array after collecting information on all minimizes, and then add it to the hash table. This procedure is claimed to be dramatically faster than direct hash table insertion and highly cache efficient[37,101,102]. Considering double strands, both forward and reverse DNA strands, of the same reference genome is needed to address the strand bias problem, which is defined as the difference in genotypes identified by reads that map to forward and reverse DNA strands[24,103].

## Pattern alignment

Modern sequencing machines still generate randomly sampled subsequences (sequencing reads) of the original genome sequence[32,104]. The resulting reads lack information about their order and corresponding locations in the complete genome. Applying a diet pattern to sequencing reads is a daunting challenge to us as the exact mapping location of each read is unknown, even with the use of read mapping tools that provide a "best guess" of mapping location given certain parameter values. Applying a pattern to a read sequence starting from the first base would provide, in most cases, a very different patterned genome sequence compared to when applying the same pattern starting from the second base (Fig. 7B). This issue becomes worse when the user's pattern sequence is not regular. To address this issue, we propose a new computational step, called pattern alignment. The pattern alignment step takes as input both a read sequence and the same user's pattern that is used for the compressed indexing step and provides as output the alignment index, $a$, which is a non-negative integer value.

The pattern alignment step exploits the following observation to calculate the alignment index: If an alignment index is appropriately chosen, then two patterned homologous regions should share a large number of seed matches. Thus, if most of the seeds collected from a patterned read sequence collectively and frequently exist in the reference genome, then the patterned read sequence is homologous to some regions in the reference genome. The pattern alignment step incrementally shifts to the right direction of the diet pattern sequence against the read sequence. The pattern alignment step generates $p - 1$ right-shifted versions of the diet pattern in addition to the original diet pattern, where $p$ is the length of the user's pattern sequence. Each of the $p$ diet pattern sequences is applied to the read sequence to obtain a patterned read $s$, $PQ_s$, where $p - 1 \geq s \geq 0$ and $s$ is the shift amount used to shift the diet pattern sequence to the right direction. $PQ_s$ is a subset that includes all non-zeros elements of the result set of multiplying the $s^{th}$ diet pattern with the query sequence, $PQ_s \subseteq \{Q[i+s].P[i\bmod p]\}_{i=0}^{q-1}$. The pattern alignment step collects a number of seeds and then calculates their double-strand $(w, k)$-minimizers. For each $PQ_s$, the pattern alignment step examines the presence of the extracted minimizers (their hash values) in the seed index and calculates their sum of occurrence frequencies. The pattern alignment greedily calculates the alignment index, $a$, based on the shift amount, $s$, of the corresponding $PQ_s$ sequence that leads to the highest total occurrence frequency of all calculated minimizers of each $PQ_s$, $\max\{\sum f(each\ minimizer\ of\ PQ_s)\}_{s=0}^{p-1}$.

Given that the number of iterations to check the total occurrence frequency equals $p$ and $p$ is a user-defined value that can be very large, the pattern alignment step can be $p$ times more expensive than typical seeding algorithms in read mapping that collect all possible minimizers from a read sequence only once. To efficiently address this drawback, we propose two effective solutions: 1) Discarding expensive minimizers and 2) Limiting the number of collected minimizers. As each seed contributes to the frequency counter individually, we discard the most frequently ($\geq$ a user-defined threshold) occurring minimizers from the frequency count to avoid dominating the frequency count and avoid misleading the pattern alignment step by giving the impression that all collected seeds from an $PQ_s$ are frequent. Discarding highly-frequent minimizers is already applied in existing state-of-the-art approaches[27,37,43]. We also limit the number of calculated minimizers in each $PQ_s$ to a user-defined threshold, which directly reduces the length of each $PQ_s$ from at most $q$ to $t$, where $q \gg t$ and $PQ_s \subseteq \{Q[i+s].P[i\bmod p]\}_{i=0}^{t-1}$. The pattern alignment step generates and considers the same number of minimizer seeds from each $PQ_s$.

## Compressed seeding

As we already calculate the alignment index in the second step of Genome-on-Diet, we now can correctly reduce the size of the read sequence by dropping some of its bases. The compressed seeding step takes a read sequence, $Q$, and alignment index, $a$, as an input and provides all locations of minimizers that are common between a read sequence and the seed index (Fig. 7C). The diet pattern is first shifted to the right direction by a shift amount that equals the alignment index, $a$. The shifted diet pattern is then applied to the read sequence to generate the patterned read, $PQ \subseteq \{Q[i+a].P[i\bmod p]\}_{i=0}^{q-1}$. The compressed seeding step collects all overlapping seeds, calculates their double-strand $(w, k)$-minimizers, and finds exact matches of

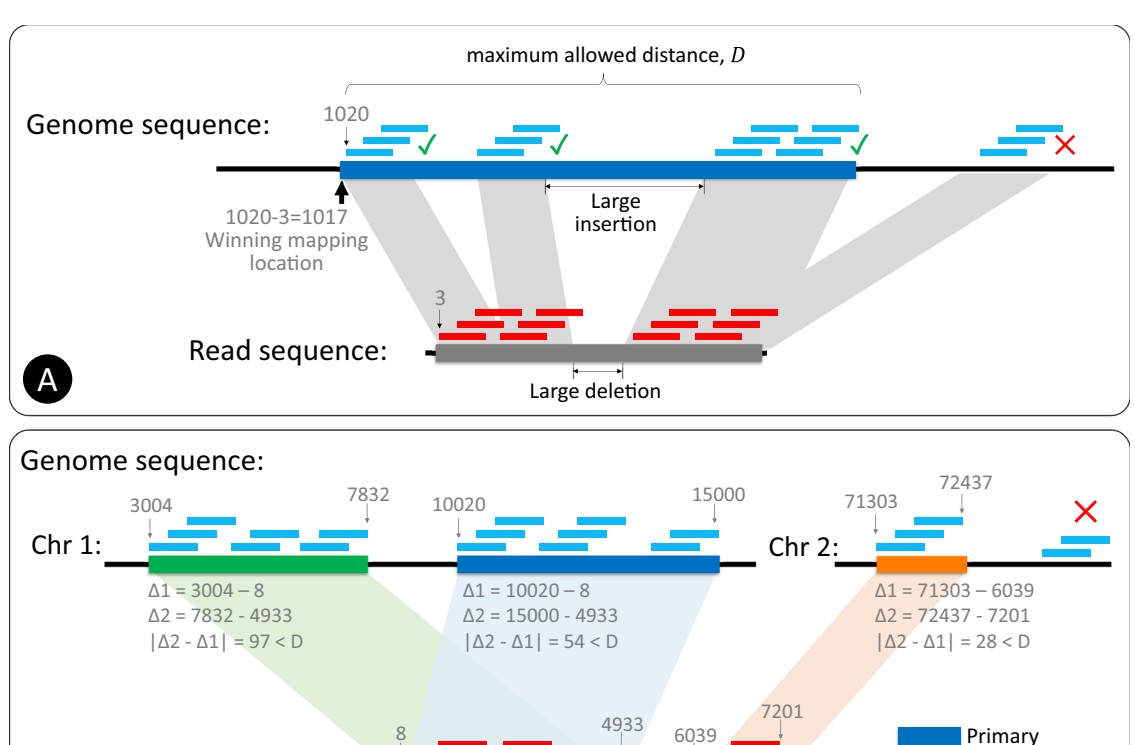

**Fig. 8 | Two different location voting algorithms for determining winning mapping locations in Genome-on-Diet. A** Location voting for Illumina reads. The location that receives the highest number of votes (i.e., location 1017 receives 12 votes) within a maximum allowed distance, $D$, is selected as the winning mapping location. Based on the ratio between the maximum allowed distance and the read length, the location voting step can consider insertions, deletions, and substitutions without any restriction on their location and their number. **B** Location voting for HiFi and ONT reads. The distance threshold, $D$, is 100 and the voting threshold, $V$, is 3. The location voting step outputs three subsequences [3004–7832], [10020–15000], and [71303–72437], which receive 7, 8, and 3 votes, respectively.

minimizers in the reference genome by querying the seed index. Other seeding approaches, such as syncmers[42,105], strobemers[106], and BLEND[54], can be used instead of the minimizer approach. However, to maintain correctness and high sensitivity, the same algorithm used to extract seeds from the patterned genome sequence must be used in the compressed seeding to calculate seeds from the patterned read sequence.

## Location voting

The location voting step takes as an input a location list of all seeds extracted from the read sequence and their matching locations in the reference genome retrieved from the seed index. The location voting step provides as an output two types of data: 1) a set of sequence pairs when the goal is to detect SNPs and indels within a read sequence as in Illumina reads where SVs are detected using spaced paired-end reads[107] or 2) a set of subsequence pairs when the goal is to detect SNPs, indels, and SVs all together within the same read sequence as in HiFi and ONT reads (Fig. 8). While the sequence pair refers to the complete read sequence and a subsequence extracted from the reference genome, the subsequence pair refers to a subsequence extracted from the reference genome and another subsequence extracted from the read sequence. The output of the location voting step is then used for performing sequence alignment. Since each minimizer seed in Genome-on-Diet does not represent exact matches (anchors[37,108]) of consecutive nucleotides in the original reference genome, methods such as seed chaining cannot be directly used to infer the locations that can lead to optimal alignments. Our location voting mechanism introduces a method that uses patterned minimizers to vote on the optimal location without the need for performing computationally

expensive calculations to examine the quality of each chain of seeds as in typical read mappers. The location voting mechanism has four main steps (besides the first three steps, we need to apply only either the fourth or the fifth step depending on the type of sequencing reads).

(1) Interpreting all matching locations based on their corresponding locations in the read sequence. We subtract the location of each seed extracted from the read sequence from that of its corresponding matching seed extracted from the reference genome. This step helps us to consider both repeated seeds (e.g., having the same hash value) that are extracted from the read sequence and repeated mapping locations only once. This prevents performing redundant computations for faster processing.

(2) Combining all lists of matching locations retrieved from the seed index into a single *sorted* list. As the seed index stores an already-sorted location list for each minimizer seed ("Compressed indexing"), we implement a branchless merge sorting[109] algorithm to obtain a single sorted list of all adjusted matching locations. We evaluate our implementation and other alternative sorting algorithms that are used in minimap2 ("Optimization strategies"). The locations in the final sorted list are already the locations of the original unpatterned reference genome, and hence, an intermediate translation between locations in the patterned genome to locations in the original genome sequence is not needed.

(3) Determining the distribution and density of locations of matching seeds in the reference genome using two steps. (1) It considers the first location in the sorted list as a temporary mapping location, $\Delta 0$. (2) It calculates the number of locations that are within a predetermined distance from the temporary mapping location (e.g., $|\Delta D - \Delta 0| \leq D$, where $D$ is a user-defined maximum allowed distance).

(4) Determining potential mapping locations for Illumina reads is performed as follows (Fig. 8A). If the total count of locations is greater than or equal to a user-defined threshold, $V$, then the mapping location $\Delta 0$ and the total number of votes are all stored in the list of winning subsequence pairs. The first mapping location that exceeds the predetermined distance, $D$, is considered as the current temporary mapping location and the location voting step repeats the previous step until reaching the last location in the list. The final list of winning subsequence pairs is sorted based on the number of votes, and sequence alignment is then performed between the read sequence and each subsequence extracted from the reference genome at each location in the winning mapping locations.

(5) Determining potential mapping locations for HiFi and ONT reads is performed differently from the fourth step (Fig. 8B). If the total count of locations is greater than or equal to a user-defined threshold, $V$, then the subsequence starting from $\Delta 0$ until $\Delta D$ in the reference genome along with its corresponding subsequence from the read sequence and the total number of votes are all stored in the list of winning subsequence pairs. The first mapping location that exceeds the predetermined distance, $D$, is considered as the current temporary mapping location and the location voting step repeats the third and fifth steps until reaching the last location in the list. The final list of winning subsequence pairs is sorted based on the number of votes and sequence alignment is then performed for each subsequence pair. For each two subsequence pairs in the list of winning pairs that are apart from each other by less than or equal to 50,000 bases, we calculate their concatenated CIGAR string that maximizes the alignment score to include large variations in the final CIGAR string. If two subsequence pairs cover different regions in the read sequence and they are apart from each other in the reference genome by more than 50,000 bases, then the pair that provides the highest alignment score is considered as primary alignment (if such one doesn't exist before for the read sequence) and the other pair is considered as supplementary alignment. In order to reduce the computation overhead of a large list of winning subsequence pairs, we limit the size of the list in both the fourth and fifth steps to a user-defined number.

Our location voting mechanism is fundamentally different from the location voting strategy of Subread[110] in four aspects. (1) Genome-on-Diet calculates several subsequence pairs for each read sequence, while Subread considers for each read sequence only a single mapping location that receives the highest number of votes. This provides two important benefits to Genome-on-Diet: supporting the detection of small and large variations (e.g., interchromosomal and intrachromosomal translocations[17]) and supporting the three main types of alignments, primary, secondary, and supplementary alignments. (2) Our location voting mechanism does not examine the quality of anchors/chains, while Subread uses Hamming distance to count the number of matches for forming chains and then performs DP-based alignment to complete the alignment between every two chains. (3) Insertions, deletions, and substitutions in our location voting mechanism are allowed to occur anywhere in the read sequence and/or the reference subsequence as Genome-on-Diet performs end-to-end sequence alignment for each sequence pair, while Subread allows them to occur only between chains as it requires performing sequence alignment only between every two chains. (4) Our location voting mechanism uses patterned minimizers of any length, while Subread uses all short overlapping unpatterned seeds.

### Sequence alignment and SAM information
After obtaining the potential subsequence pairs, Genome-on-Diet performs base-level sequence alignment for each sequence pair. We perform sequence alignment using a recent vectorized implementation[33] of KSW2[37]. Depending on the read length, we use two different strategies for performing sequence alignment. For short reads, we perform global alignment, where the entire read sequence is aligned against the genome subsequence. For long reads, we only perform local alignment, where the query subsequence is aligned against the reference subsequence. We observe that the voting distance is directly affected by the number of consecutive insertions or deletions, causing the matching seeds to be apart from each other or scattered into different regions. Hence, the voting distance value can be used as the width of the band in banded alignment. This insight helps to optimize the computation time and the memory consumption of the sequence alignment step without losing accuracy. Unlike minimap2, Genome-on-Diet performs sequence alignment for the complete sequence pair and not only between adjacent anchors in a chain. The approach of minimap2 is known as sparse DP, which can provide suboptimal alignment calculation as demonstrated by the authors in[111].

Sequence alignment algorithms use computationally expensive DP[100,112–115] algorithm to optimally (1) examine all possible prefixes of two sequences and track the prefixes that provide the highest possible alignment *score* (known as optimal alignment), (2) identify the type of each difference (i.e., insertion, deletion, or substitution), and (3) locate each difference in one of the two given sequences. Such alignment information is typically output by read mapping into a sequence alignment/map (SAM, and its compressed representation, BAM) file[75]. The alignment score is a quantitative representation of the quality of aligning each base of one sequence to a base from the other sequence. It is calculated as the sum of the scores of all differences and matches along the alignment implied by a user-defined scoring function. A mapped read can have multiple mapping locations. The SAM file usually contains 4 types of records, primary, supplementary, secondary, and unmapped. In Genome-on-Diet, the mapping location that leads to the highest alignment score is considered the primary alignment. If another subsequence pair for the same read sequence is shorter than 80% of the length of the subsequence pair assigned as primary alignment, then the record is considered as supplementary alignment. Otherwise secondary alignment.

DP-based approaches usually have quadratic time and space complexity (i.e., $(q^2)$ for a query length of $q$), but they avoid re-examining the same prefixes many times by storing the examination results in a DP table. The use of DP-based approaches is unavoidable when optimality of the alignment results is desired[113]. We refer the reader to comprehensive surveys[24,31,32] of acceleration efforts for improving the sequence alignment step using algorithms and hardware accelerators.

Recent read mappers usually report a mapping quality (MAPQ) value per each primary alignment, a measure of the confidence that a read actually comes from the mapping location it is aligned to by the mapping algorithm[116]. Genome-on-Diet provides an empirical MAPQ calculation that is guided by that of minimap2 (Supplementary Table 12).

### Optimization strategies
We introduce four different optimization strategies that improve the overall performance and/or accuracy of one or multiple steps of Genome-on-Diet. The four optimization strategies can be conveniently configured, enabled, and disabled using input parameter values entered in the command line of Genome-on-Diet.

**Accelerating seeding with SIMD instructions.** We observe that extracting seeds from both reference genome sequence and read sequence is performed sequentially. That is, the second seed will not be extracted before extracting the first seed. Thus, seed extraction is performed in linear time with regard to the subject sequence length, which can be a few billion bases long. Seed extraction in Genome-on-Diet is already a computationally critical step as it is used in three main steps: (1) Compressed indexing, (2) Pattern alignment, and (3) Compressed seeding.

To further accelerate these three key steps, we introduce a vectorized implementation for each of these three steps. Each implementation provides a holistic acceleration starting right after applying the pattern to the subject sequence. Thus, in the three steps of Genome-on-Diet, we (1) vectorize the implementation for extracting overlapping k-mers, (2) calculate the hash value of each k-mer, and (3) find the k-mer with the minimum hash value. The key idea of our implementation is to process every 8 overlapping k-mers in parallel during a single iteration. This restricts the length of each k-mer sequence to only 32 bases (assuming 512-bit wide SIMD registers and 2-bit encoding for each DNA base), which is the same restriction as minimap2 (k is up to 28 bases, and another 8 bits are used for storing other seed information).

During any iteration, we read 8 additional consecutive bases from the subject sequence and encode each base using 2 bits. The 8 new bases in addition to k prior bases (extracted already in prior iterations) will be used to compute 8 new k-mers. We calculate the hash values of all 8 k-mers at once using our vectorized implementation of Thomas Wang's hash function[117]. We also compute the minimum hash value among the 8 hash values at once. Depending on the size, w, of the minimizer window, we keep searching for the single k-mer that has the minimum hash value among w overlapping k-mers by processing the k-mers in groups of 8 k-mers. If an ambiguous base (N) is encountered, we terminate the current computation and search for the next minimizer seed starting from the base that is located right after the ambiguous base. This helps us to use only 2-bit encoding for each DNA base, instead of 4-bit encoding. Handling ambiguous bases in this way induces a non-negligible cost of terminating the loop and wasting the already-computed vectors, which is unavoidable as our goal is to maintain the exact same approach of minimizer seed extraction in minimap2.

Our vectorized and non-vectorized implementations are strictly equivalent. That is, for the same input sequence and the same input parameters, both implementations return the same list of minimizers. Our holistic acceleration approach provides benefits to the end-to-end execution time of seed extraction. We tested our implementation for different k and w values (Supplementary Table 13). Our implementation uses the Intel AVX-512 extension that is supported by most modern Intel CPUs (Intel Skylake generation and successors). We observe that our implementation provides up to 1.66× speedup for small values of w and up to 2.24× speedup for large values of w (Supplementary Table 13). We observe that the vectorized implementation is slower than the non-vectorized implementation when the input patterned sequence is *very short* due to the overhead of loading the AVX registers. Users can still choose between the vectorized and the non-vectorized implementation for supporting a wide range of processors.

**Sorting seed locations.** In the location voting step of Genome-on-Diet, we require sorting the seed locations retrieved from querying the compressed index before performing location voting. The retrieved location list per query seed can be very large, depending on the length of the reference genome and parameter values used for seed extraction. The number of location lists that need to be sorted per read is up to the number of extracted seeds from the read sequence. Sorting multiple location lists per read is computationally expensive, especially when the number of seeds extracted from a read sequence is large. We observe that each hash value stored in the compressed index has its own list of locations that will be retrieved when the query seed has the same hash value. Each location list is already sorted, and hence, we believe that merge sort can be beneficial. Hence, we propose using merge sort as opposed to using radix sort (for long reads) or heap sort (for short reads) as in minimap2 to sort the location list of matching seeds.

We assess the benefit of using merge sort over radix sort and heap sort for sorting retrieved locations (Supplementary Table 14). We observe that for a short list of locations, merge sort provides the fastest performance, while for a long list of locations radix sort is the fastest. However, the performance of merge sort is still comparable to that of radix sort for a large number of locations. Thus, we choose merge sort as the default sorting algorithm as it works well for different data types and different sizes of the location list. Out of the three sorting implementations, only radix sort provides an in-place sorting, and hence, it shows the least memory footprint (Supplementary Table 14). Users are still able to choose between the three types of sorting algorithms.

**Rescuing mapping location.** In the location voting step of Genome-on-Diet, a read could have very few (less than a user-defined threshold V) matching minimizers for various reasons. To rescue at least a single mapping location for such a read and maintain high sensitivity (mapping as many reads as possible), we always store a single rescue mapping location per read that has the highest number of votes regardless of whether the number of votes is less than the user-defined threshold, V. In case the list of winning mapping locations is empty, then the location voting step provides the rescue mapping location as output. Although such rescued alignments increase the overall execution time of Genome-on-Diet as they require performing sequence alignment, they improve sensitivity with no additional overhead provided by the rescuing mapping location step.

**Handling exactly-matching short reads.** Performing sequence alignment is still computationally expensive, and it is an open research problem[100,112–115,118]. Due to the low sequencing error rates of Illumina sequencing machines, it is observed that a large fraction of short reads typically maps exactly or with a few mismatches to the reference genome[119–122]. For example, on average 80% of human short reads map exactly to the human reference genome[119]. We employ a quick filter[121] that detects exactly matching reads using SIMD instructions and outputs their alignment information directly to the SAM file without performing sequence alignment calculations for such reads.

### Reporting summary

Further information on research design is available in the Nature Portfolio Reporting Summary linked to this article.

## Data availability

This study used publicly available data for evaluation. The short reads (Illumina), the accurate long reads (HiFi), and the ultra-long reads (ONT) are obtained from the NIST's GIAB project: https://ftp-trace. ncbi.nlm.nih.gov/ReferenceSamples/giab/data/AshkenazimTrio/ HG002_NA24385_son/. We provide the exact sources and direct links to the sequencing read sets in Supplementary Table 1. The complete Human Genome GRCh38 can be accessed using https://ftp-trace.ncbi. nlm.nih.gov/ReferenceSamples/giab/release/references/GRCh38/ GCA_000001405.15_GRCh38_no_alt_analysis_set.fasta.gz. We use wgsim (v1.0 conda) to simulate the reads used to produce Fig. 6. The wgsim command lines are provided in: https://github.com/CMU-SAFARI/Genome-on-Diet/blob/main/ReproducibleEvaluation/ ContainmentSearch/ContainmentSimulation.sh. All other data supporting the findings described in this manuscript are available in the article and its Supplementary Information files.

## Code availability

Genome-on-Diet code is available at: https://github.com/CMU-SAFARI/ Genome-on-Diet. The exact release of the source code used in this work is https://doi.org/10.5281/zenodo.13674286.

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

## Acknowledgements

We thank Can Alkan (Bilkent University), Heng Li (Harvard University), and the organizers and attendees of the International Genome Graph Symposium 2022, Switzerland (https://iggsy.org) for their valuable feedback and discussion. O.M. acknowledges the generous donation support provided by the industrial partners of the SAFARI Research Group that has indirectly contributed to the positive environment that enabled this research. This work was supported and enabled by the endowment of O.M.'s professorship provided by ETH.

## Author contributions

M.A. conceived of the presented idea. M.A. and J.E. implemented the tool. All authors wrote, reviewed, and edited the manuscript. All authors discussed the text and commented on the manuscript. All authors read and approved the final manuscript. O.M. supervised the study.

## Funding

## Competing interests

All authors declare no competing interests.
