## [Transparent Peer Review file · Nature Communications]

Taming Large-Scale Genomic Analyses via Sparsified Genomics

Corresponding Author: Dr Mohammed Alser

Version 0:

Reviewer comments:

Reviewer #1

(Remarks to the Author)

In the manuscript submitted by Alser et al. to the journal nature communications the authors present a new algorithm, Sparsified Genomics, that can be utilized to decrease overall computational load in the scope of read mapping. Overall, the presented manuscript is extensive and the authors have put great effort in showing the benefits that their proposed method exhibits throughout three major evaluations. Even so, it is not entirely clear what major benefit sparsified genomics provides as a whole or what achievement is considered to be the most relevant. Consequently, several major issues will be raised hereafter.

Major point 1: The authors have carried out a series of evaluations (Chapter 2.1, 2.2, and 2.3) on their tool and presented its superior performances in terms of computational runtime, memory consumption, accuracy, sensitivity, etc. However, a very critical drawback is that each benchmarking is carried out independently on a conditioned data set and for each data set different (rather than all) performance metrics are presented. A significant runtime improvement can not be justified without showing a decent sensitivity and accuracy at the same dataset. This kind of evaluation can not present the whole picture of the performance of the tool. I suggest the authors to:

Adding a comprehensive evaluation to present all aspects/metrics on the same sets of data and present all metrics to see if there is any kind of trade-off in performances by introducing sparsified genome.

Or include all missing metrics for the evaluations on all the datasets.

Major point 2: there should be more investigations on how different patterns could impact the outcome of the mapping results. A noteworthy question would be, what are the best patterns to choose when encountering different application scenarios. This also raises the question, why for the evaluation that also focuses on sensitivity (Chapter 2.1.) only the pattern "10" was used, while for the following evaluation focused solely on runtime more complex patterns, e.g. "100" or "1110", were used.

Major point 3: The explain (at the end of Chapter 1.2.) that using sparsified genomics already on genome or read level rather than on seed level (as e.g. carried out by spaced seeding) will eventually result in differently patterned seeds (in dependence on the actual pattern) and by that "improve the overall sensitivity as it allows for tolerating variations of different types, quantities, and locations". However, this claim is pure speculation as is not backed up by any profound evaluation or a relevant reference on this exact matter. Even more, the effect of complex patterns on the sensitivity and specificity warrants a more in depth analysis or at least discussion. For instance, is there a possibility that a specific pattern can also reduce sensitivity if it is applied to regions with (tandem) repeats, for instance, that occasionally coincide with the repeating pattern?

Major point 4: In Chapter 2.1.3. the authors conclude that "Genome-on-Diet is [...] of high read mapping quality and accuracy". First of all, accuracy was not measured as accuracy is defined as the ratio between all TPs + TNs and the total number of all true and false cases (and TN values are not defined/provided). Even if the authors define "accuracy as sensitivity" (see Chapter 1, page 4, paragraph (3)) this is misleading, in particular in this case as this suggests a low FP rate with the already shown low FN rate. Therefore I would suggest not to redefine the usual and known designations for those statistics. In addition, the authors should also, at least, include the fact that for this part of their evaluation the FP-rate for SNPs was always higher for Genome-on-Diet compared to minimap2 (for reference see Figure 2 J, K, L). In addition, this aspect could also be more discussed by the authors, i.e. the ever present tradeoff between precision and recall and what of those Genome-on-Diet is trying to maximize and why?

Major point 5: At the end of Chapter 2.2.2. the authors conclude that Genome-on-Diet is maximally beneficial for very large-scale analysis with resource-limited infrastructure. This, again, is somewhat exaggerated. First of all, very large-scale genomics becomes interesting in the range of terabytes not gigabytes as used here. So the question would be to what extent Genome-on-Diet could tone down e.g. the memory requirement if using the complete RefSeq database, or at least about 10-25%, while also providing meaningful results? Secondly, the authors argue that the RefSeq database approximately doubles every 3 years (page 12, Chapter 2.2.), which corresponds to an exponential increase. However, their improvement on speed and memory preservation is clearly shown to be linear, i.e. using the pattern "110" (that is essentially two third of the input data) in comparison to "100" (that is one third of the input data) results in a linear improvement (one third to two third). How is Genome-on-Diet proposed to counter the exponential data explosion then?

Major point 6: For their evaluation on taxonomic profiling in Chapter 2.3.1. the authors describe that they have disabled the recovery mode for this part. To our understanding this feature is, however, one of the algorithmic specifications that is responsible for sustaining a high sensitivity. Controversy, disabling this feature could result in an increased FN-rate, or if all algorithms perform equally well on this regard anyway (see Chapter 2.3.1.), enabling this feature could result in an increased FP-rate. Is this also a reason for the differences on the L1-norm-error and to what extent does this influence the overall performance?

Major point 7: In Chapter 2.3.1 (first paragraph) the authors argue that the higher memory footprint of Genome-on-Diet in comparison to Metalign can be counteracted by reducing the batch size. However, taking the results shown in Table 2, a reduced batch size comes at the cost of an increased run time. This conflicts with the overall goal and claim that Genome-on-Diet is superior in terms of both memory requirements and runtime.

Minor point 1: In the PAF format output by the tool, the query's starting location on the reference is 1 index lower, resulting in the start to end position longer than the read length itself.

Minor point 2: The order of the Figures and Tables should match their occurrence in the main manuscript (e.g. see page for where Table 4 is referred even before Table one). I suggest to relabel or rearrange this accordingly.

Minor point 3: The authors write that Genome-on-Diet provides a lossless compression of genomic sequences. To my understanding this is an exaggeration or at least somewhat misleading. If the original genomic sequence needs to be retained (as only a copy of the original genomic sequence is sparsified) in order to reverse the process losslessly it is hardly a compression, even if the sequences are compressed by other means (i.e. using gzip). At any rate, compression is not the application scenario for this newly introduced method. Therefore, I argue against raising such expectations unless it is shown by a proper evaluation or proven methodologically.

Minor point 4: In Chapter 2.2.2. (page 14) the authors explain that "Genome-On-Diet" requires carefully adjusting the thresholds for the location voting step when using different pattern sequences". What are the recommendations here if any other than the default pattern is used? What were the precise parameters for all parts of the three different evaluations? Does this parameter also need to be adjusted for different input data or just for different patterns?

Reviewer #2

(Remarks to the Author)

The manuscript by Alser et al. presents a novel strategy to deal with the continuously increasing amount of genomic data and the very mapping analysis tasks that are commonly done using this data.

The manuscript is well-written and organized, although there are a few points that should be considered and revised, according to this reviewer's opinion.

One is from the title, after reading through the manuscript, the "Genomic Analyses" words in the title might be too broad. For instance, analyses such as gene calling and how will they be affected by the use of this approach. Is in this reviewer's opinion that this approach can be very useful, as presented in the manuscript, for mapping tasks, but other genomic analyses might not be performed the same as when using the full genomic data. One suggestion for revising the title could be "Taming Large-Scale Mapping Analyses via Sparsified Genomics", but others the authors might think will better give a grasp of their approach could work as well.

Another aspect that is stressed throughout the manuscript is the reduced size of the database by employing this approach. This is in general true if you only consider the database, but to be honest the full genomic data must be stored and present somewhere (which might not be true for other mappers for which the full sequence can be directly reconstructed from the indexed database). Hence, this should be made clear up-front and maybe not stressed that much. As if someone will need to change the binary pattern would still need to go back to the full genomic data.

A third important aspect is probably not stressed enough. Is understandable that is difficult to collect and present all aspects in a very detailed manner. However, the choice of the binary pattern (or 'diet pattern') seems to be very crucial for having comparable performances when using the Genome-on-Diet framework. So, practically, to make this easily adopted by the community, it will be very important to be able to provide some examples about the effect of different choices for this pattern and how this can also affect performances.

Lastly, a minor point, but since bowtie2 is also a very common mapping tool, it will be very nice by the authors if they could include it as well in their comparisons.

Version 1:

Reviewer comments:

Reviewer #2

(Remarks to the Author)

This reviewer thanks the authors for the revised version of the manuscript. All points were addressed and the newly added information and comparison improved the overall quality and accessibility of the manuscript.

Reviewer #3

(Remarks to the Author)

Comments

The authors appropriately addressed most of the comments raised in the previous review round, also providing additional material.

However, in my opinion, among the newly provided material, the results presented in section "2.1.1 Evaluation methodology" and "2.1.2 Sparsified genomics allow for tolerant variations" are not completely fair. I list below the main concerns:

1) the patterns chosen for Genome-on-Diet are among those that best perform (according to the results shown in subsequent sections). However, it is not clear how the parameters were chosen for the other pattern schemes to which they are compared.

For all-seed (which I understand are basically k-mers) a length 8 is chosen. Why is it so? Did the authors tried several lengths and showed the best results? If so this should be made clear in the manuscript. The same holds for minimizers: size 6 in a window of length 8 is the best performing combination for minimizers or a randomly chose one? As for spaced-seeds I understand the authors wanted to test the same 0-1 patterns used for genome-on-diet, but this is not fair because, as the authors explained themselves, spaced-seeds work in a very different manner, much closer to how k-mers work (but including of course wildcards). I will detailed several specific observations about spaced-seeds in the following comments.

2) In all the literature that I am aware of, spaced-seeds are defined as patterns of 0s and 1s that always begin and end with a 1. According to this definition, 3 out of 4 of the tested patterns are not technically spaced-seeds.

I point out to the authors, as a valuable source of information about spaced-seeds, the web page:

<https://sites.google.com/view/laurentnoe/spaced-seeds>

Although I see it is no longer maintained, the literature list is updated up to 2020, so it has all the references related to basics definitions.

3) The length of the spaced-seed mask that was tested is limited to 2 or 3 nucleotides for 3 out of 4 of the tested spaced-seed masks. This length is definitely too short for sequences of length 1000. If $k=8$ is indeed the best length for the all-seed scheme, then I guess reasonable combination of length-weight for spaced-seeds (where with weight I mean the number of 1s) could be 8-6 or 10-8. In any case, this is just a guess and proper experiment should be done. Since for genome-on-diet the best pattern were shown, also some test with different parameters for the other schemes should be carried out.

4) In real applications the positions of the 1s in a spaced-seeds mask is not random, but the result of the maximization (or minimization) of an objective function (e.g. minimizing the overlap or maximizing the sensitivity). There are tools that can compute the best distribution of 1s and 0s, e.g. rasbhari:

L. Hahn, C.-A. Leimeister, R. Ounit, S. Lonardi, and B. Morgenstern, "rasbhari: optimizing spaced seeds for database searching, read mapping and alignment-free sequence comparison," PLoS Computational Biology, vol. 12, p. e1005107, October 2016.

Other examples of "good" spaced-seeds can be found in the paper related to the target application of interests.

Version 2:

Reviewer comments:

Reviewer #3

(Remarks to the Author)

The authors added some experiments using spaced seeds masks that are more appropriate for spaced seeds, as per previous reviewers comment. As expected, with these spaced seed the performances of the "spaced approach" improves making the comparison with the proposed sparsified approach fairer, without affecting the overall result of the proposed approach.

I have no further major requests, I just spotted a couple of typos to fix:

- 1) Page 12, second line from the bottom
Bowttie2 -> Bowtie2
- 2) Reference 76 is incomplete: authors are missing

Reviewer #1 (Remarks to the Author):

Comment 1: In the manuscript submitted by Alser et al. to the journal nature communications the authors present a new algorithm, Sparsified Genomics, that can be utilized to decrease overall computational load in the scope of read mapping. Overall, the presented manuscript is extensive and the authors have put great effort in showing the benefits that their proposed method exhibits throughout three major evaluations. Even so, it is not entirely clear what major benefit sparsified genomics provides as a whole or what achievement is considered to be the most relevant. Consequently, several major issues will be raised hereafter.

Response 1: We thank the reviewer for finding our manuscript extensive and acknowledging that there is great effort invested throughout the three major evaluations.

To address the comment of the reviewer, we added a new paragraph to Section 1.1 to summarize the major benefits and drawbacks of sparsified genomics as highlighted below.

Major Benefits and Downsides of Sparsified Genomics. Sparsified genomics provides in principle several key benefits: 1) Reduced total execution time due to processing less workload (smaller number of included bases), 2) Reduced peak memory footprint due to smaller number of extracted seeds and hence a smaller index, 3) No need to pre-build genome indexes as it is feasible to build it during the analysis with low performance-overhead. The seeds in sparsified genomics are designed to tolerate more mismatches per read sequence depending on the number of zeros used in the pattern sequence. This might lead to finding a large number of sequences in the reference genome that are similar to the given read sequence and hence detecting more genomic variations and possibly some falsely detected variations. We experimentally evaluate and quantify these benefits in detail in the next section.

Comment 2: Major point 1: The authors have carried out a series of evaluations (Chapter 2.1, 2.2, and 2.3) on their tool and presented its superior performances in terms of computational runtime, memory consumption, accuracy, sensitivity, etc. However, a very critical drawback is that each benchmarking is carried out independently on a conditioned data set and for each data set different (rather than all) performance metrics are presented. A significant runtime improvement can not be justified without showing a decent sensitivity and accuracy at the same dataset. This kind of evaluation can not present the whole picture of the performance of the tool. I suggest the authors to:

Adding a comprehensive evaluation to present all aspects/metrics on the same sets of data and present all metrics to see if there is any kind of trade-off in performances by introducing sparsified genome.

Or include all missing metrics for the evaluations on all the datasets.

Response 2: We, unfortunately, do not agree with the reviewer's comment. The three aforementioned evaluations (Chapters 2.2, 2.3, and 2.4 in the updated manuscript) are targeting totally different key applications, read mapping, containment search, and metagenomics profiling. Each of these applications has its own well-established benchmarking tools, performance metrics, and ground-truth datasets. Throughout the evaluation for the same application, we always use the exact same data for both Genome-on-Diet and all other baselines. For example, in read mapping, we used three state-of-the-art read sets (from GIAB consortium as we discuss in Supplementary Table 1) to evaluate both Genome-on-Diet and the best-performing read mapper, minimap2. We use the first read set (using Illumina sequencing technology) in Figures 4A,D,G,J to evaluate both Genome-on-Diet and minimap2.

We use the second read set (using HiFi from PacBio sequencing technology) in Figures 4B,E,H,K to evaluate both Genome-on-Diet and minimap2. We use the third read set (using nanopore sequencing technology) in Figures 4C,F,I,L to evaluate both Genome-on-Diet and minimap2. We also use a state-of-the-art sensitivity and accuracy analysis for the tools running the exact same dataset. Our sensitivity analysis for read mapping uses the most recent and comprehensive benchmark for genetic variations (<https://www.nature.com/articles/s41587-021-01158-1>). More details on the sensitivity/accuracy analysis are provided in Section 2.2.1. Evaluation Methodology.

We, unfortunately, cannot use the same datasets for the three aforementioned applications (read mapping, containment search, and taxonomy profiling). For example, the accuracy of read mapping is normally determined by the ability to detect true variations, while the accuracy of taxonomy profiling is determined by the ability to calculate 1) correctly the presence/absence of all species and 2) accurately the relative abundance of each species. If we use for read mapping evaluation the same read set (CAMI datasets) we used in taxonomy profiling evaluation, then we won't be able to benchmark genetic variation detection as there are no ground-truth variations for such read sets (CAMI datasets). In each application evaluation, we already used state-of-the-art baselines, well-studied read sets, different genomes, and best-practice benchmarking tools for the three applications as we mention in the evaluation methodology section of each corresponding application.

However, in case the reviewer is interested in seeing the performance of the three applications when using the exact same read set, the third application (taxonomy profiling) already followed such a scenario (as we explain in the second paragraph in Section 2.4. Taxonomic Profiling) where it uses KMC3+CMash (containment search application) as the first step then minimap2 (read mapping application) as the second step. We followed the same approach in our evaluation of Genome-on-Diet for taxonomy profiling (more details in Section 2.4.1. Evaluation Methodology). We provided the results in Tables 6 and 7. Please note that in Table 7, we used the best (both comprehensive and accurate) benchmarking tool, called OPAL, which is widely used in academia and recommended by best-practice of the CAMI challenge consortium (<https://www.nature.com/articles/s41592-022-01431-4>).

Comment 3: Major point 2: there should be more investigations on how different patterns could impact the outcome of the mapping results. A noteworthy question would be, what are the best patterns to choose when encountering different application scenarios. This also raises the question, why for the evaluation that also focuses on sensitivity (Chapter 2.1.) only the pattern “10” was used, while for the following evaluation focused solely on runtime more complex patterns, e.g. “100” or “1110”, were used.

Response 3: We apologize for not including such an important evaluation. We agree with the reviewer on the need for evaluating the effect of using different patterns. We now added several new experiments where we evaluated 12 different pattern sequences for short read mapping and 3 more pattern sequences for long read mapping. We describe the new experiments and the resulting observations in Section 2.2.2 (the mentioned Chapter 2.1 by the reviewer is now Section 2.2 in the updated manuscript) as shown below. We also find out that the pattern ‘10’ provides the best accuracy-speedup tradeoff compared to that of all evaluated pattern sequences.

2.2.2. Sparsifying Genomic Sequences Significantly Accelerates Read Mapping

The first question we need to answer is which pattern sequence a user can choose for sparsified read mapping. To answer this key question we evaluate the effect of using different pattern sequences on the detected indels and SNPs and the execution time of Genome-on-Diet (**Table 2**). We make five key observations: (1) Compared to non-sparsified read mapping (using Genome-on-Diet with a pattern of '11'), sparsified read mapping can increase the number of correctly detected genomic variations as sparsified read mapping allows tolerating more differences when performing seed matching. (2) The use of pattern '10' provides the best performance when considering all evaluated metrics collectively compared to all evaluated pattern sequences, which is expected based on our analysis (**Section 2.1.2**). The use of pattern '10' increases the number of correctly detected variations by 4% and decreases both the number of missed variations and the execution time of read mapping by 25.9% and 28.4%, respectively, compared to that provided by the non-sparsified read mapping. (3) Regardless of the pattern used, sparsified read mapping comes with the drawback of increasing the number of incorrectly detected variations. This is because of the ability of Genome-on-Diet to tolerate more differences between seeds compared to other methods (e.g., minimizer seeds) that use non-sparsified (contiguous) seeds. This leads to detecting more mapping locations per read and hence more variations (both true and false) to be detected. This can be possibly addressed by applying quality filtering mechanisms that, for example, examine the number of matching bases within the matching seeds instead of only quantifying the matching seeds (as in our location voting step) since the seeds are sparsified. (4) The location of zeros in the pattern sequence has a slight effect on all evaluated metrics. That is, the patterns '110', '101', and '011' (similarly patterns '101001', '100101', '001101') all provide similar performance. However, the number of zeros in the pattern sequence has a significant effect on all evaluated metrics. (5) Though both '10' and '101001' patterns lead to excluding half of the bases from the reference genome and read sequences, they lead to different read mapping performances. This is mainly because each pattern may result in applying different patterns to each seed depending on the location of the extracted seed in the diet pattern. For example, the pattern '10' results in applying one of the two patterns, '101010...' or '01010...', to each overlapping seed, while the pattern '101001' results in applying one of the three patterns, '1010011...', '01001101...', or '001101001...', to each overlapping seed (**Figure 3c**). This directly affects the number of potential mapping locations that need to be examined, which in turn can affect the mapping results.

We conclude that the use of pattern '10' still provides the best accuracy-speedup tradeoffs for sparsified short read mapping. We also make the same observation for long read mapping (**Table 3**). Thus, we decide to use the pattern '10' for the read mapping application.

Table 2: Number of (correctly detected, incorrectly detected, and missed) indels/SNPs and total execution time as provided by Genome-on-Diet using different pattern sequences and Illumina preset (k=21 and w=11).

Pattern Sequence	INDELS / SNPs			Execution time (sec)
	Correctly detected	Incorrectly detected	Missed	
11	18'362	86	2'862	12'814.20
110	19'099	963	2'125	11'129.16
101	19'023	1'066	2'201	11'116.24
011	19'070	881	2'154	11'034.62

10	19'105	787	2'119	9'178.78
1001	18'982	1'370	2'242	9'130.61
100	19'025	1'462	2'199	8'086.17
101001	18'067	2'531	3'157	6'351.91
100101	18'194	2'463	3'030	6'565.39
001101	18'087	2'736	3'137	6'499.64
1111100000	18'840	1'763	2'384	9'926.99
1001010011	18'811	1'740	2'413	9'936.54

Table 3: Number of (correctly detected, incorrectly detected, and missed) indels/SNPs and SVs along with the total execution time as provided by Genome-on-Diet using different pattern sequences and HiFi preset (k=19 and w=19).

Pattern Sequence	INDELS / SNPs			SVs			Execution time (sec)
	Correctly detected	Incorrectly detected	Missed	Correctly detected	Incorrectly detected	Missed	
11	16'731	1'964	4'493	180	11	36	42'494.66
10	17'795	2'028	3'429	193	57	23	37'787.72
100	14'042	1'822	7'182	141	57	75	15'080.8

Comment 4: Major point 3: The explain (at the end of Chapter 1.2.) that using sparsified genomics already on genome or read level rather than on seed level (as e.g. carried out by spaced seeding) will eventually result in differently patterned seeds (in dependence on the actual pattern) and by that “improve the overall sensitivity as it allows for tolerating variations of different types, quantities, and locations”. However, this claim is pure speculation as is not backed up by any profound evaluation or a relevant reference on this exact matter. Even more, the effect of complex patterns on the sensitivity and specificity warrants a more in depth analysis or at least discussion. For instance, is there a possibility that a specific pattern can also reduce sensitivity if it is applied to regions with (tandem) repeats, for instance, that occasionally coincide with the repeating pattern?

Response 4: We thank the reviewer for this interesting comment that we repeatedly received when presenting our work. We now clarify and exemplify the key differences between spaced seeding and Genome-on-Diet using Figure 3 and the new Section 2.1 (the mentioned Chapter 1.2 by the reviewer is now Section 2.1 in the updated manuscript). We experimentally evaluate in the new Section 2.1.2 and Figure 4 the differences between extracting all possible overlapping seeds, extracting minimizer seeds, extracting spaced seeds, and extracting Genome-on-Diet seeds for different pattern sequences. We provide the new changes highlighted in yellow as follows:

2.1. Sparsified Genomics is a Unique Novel Mechanism

To our knowledge, this work is the *first* to introduce the concept of sparsifying genomic sequences and processing sparsified sequences in a very fast, efficient, and accurate way. Genome-on-Diet is *fundamentally different* from other techniques (e.g., spaced seeds⁴⁶⁻⁵¹) that apply patterns to genomic sequences in five important aspects. (1) Genome-on-Diet applies a repeating pattern to the genomic sequence (i.e., reference genome or sequencing read), while spaced seeding keeps the genomic sequence unchanged and applies a pattern to each extracted seed (**Figure 3**). (2) The resulting seed in Genome-on-Diet spans a much wider region in the reference genome compared to spaced seeds, which is of vital importance for containment search and metagenomics applications. (3) Genome-on-Diet can use a pattern sequence of any length, while spaced seeding has to use a pattern sequence of length equal to the seed length. (4) Genome-on-Diet avoids extracting seeds that start at a location overlapping with a corresponding 0 in the pattern sequence, while spaced seeding extracts the same number of seeds regardless of the pattern used. For example, using a pattern of '1000101001' results in extracting *only* 3 overlapping seeds in Genome-on-Diet, while spaced seeding extracts 6 overlapping seeds from the same region (**Figure 3**). (5) The resulting seeds in Genome-on-Diet are eventually formed based on different patterns depending on the pattern sequence and the location of the seed with respect to the repeating pattern, while spaced seeding usually applies the same pattern to all seeds. For example, with a pattern of 1000101001, Genome-on-Diet forms the first three seeds with patterns of '1000101001', '0001010011', and '0100110001', respectively, while spaced seeding forms all seeds with a pattern of '1000101001' (**Figure 3**). We also provide other examples of how a single user-defined pattern leads to applying different patterns to each seed (**Figure 3c**). We experimentally investigate the benefits and downsides of the differences between Genome-on-Diet and spaced seeding.

Figure 3: Example of resulting seeds after applying (a) spaced seeding and (b) Genome-on-Diet on a given genome sequence using the same user-defined pattern. (c) Examples of how each overlapping seed may have its own pattern. The examples at the top, middle, and bottom use user-defined patterns of '10', '101', and '101001', respectively.

2.1.1. Evaluation Methodology

Unfortunately, we are not aware of any recent read mapper that uses spaced seeding and can compete with the state-of-the-art read mapper, minimap2. To be fair with spaced seeding, we build a computer program that 1) takes a genomic sequence, 2) introduces a copy of the input sequence with a random number of substitutions at random locations such that we evaluate sequence pairs with a wide range of locations and number of differences, 3) extracts genomic seeds from the input sequence and stores them in a list, 4) extracts genomic seeds from the mutated sequence and quantify how many of these seeds appear in the list of stored seeds, and 5) repeats the first four steps for every input genomic sequence. Our computer program uses four different algorithms to extract seeds from the same sequence pair: all overlapping seeds (“All Seeds” in **Figure 4**), minimizer seeds (“Minimizers”), spaced seeds (“Spaced”), and Genome-on-Diet seeds (“Genome-on-Diet”). We use “All Seeds” and “Minimizers” algorithms as a reference for the number of matching seeds. It outputs the Levenshtein distance⁷¹ for each sequence pair along with the seed matching rates calculated when using each of the four seeding algorithms. We empirically run our computer program using a seed length of 8 and a minimizer window of size 6 and using 1000-long sequences. We use four different pattern sequences, '110', '10', '101001', and '100'. We make our computer program publicly available through the same GitHub project of this work.

2.1.2. Sparsified Genomics Allows For Tolerating Variations

We evaluate the effect of using multiple different patterns for extracted seeds as in Genome-on-Diet compared to using the same pattern sequence for all extracted seeds as in spaced seeding (**Figure 4**). We use two performance metrics: 1) *Seed matching rate*, which we define as the ratio of the number of matching seeds (i.e., seeds extracted from one sequence that match the seeds extracted from the other sequence) to the total number of extracted seeds. Ideally, the higher the seed matching rate between two sequences, the higher the similarity between the two sequences. Thus, a well-performing seeding algorithm should provide an inversely proportional relationship between the seed matching rate and edit distance. 2) *Number of accepted sequence pairs*, which we define as the number of sequence pairs that have a seed matching rate greater than or equal to a seed matching rate threshold. We determine the seed matching rate threshold as the minimum seed matching rate for all sequence pairs that have an edit distance less than or equal to a specific edit distance threshold. For example, If we want to accept all sequence pairs that have an edit distance threshold of 20, we calculate both the edit distance and the seed matching rate for all input sequence pairs and we consider the minimum seed matching rate of all sequence pairs that have an edit distance of at most 20 as the target seed matching rate threshold.

We make two observations. (1) “Genome-on-Diet” shows higher distinguishability than “Spaced” between sequence pairs with low edit distance and sequence pairs with high edit distance (**Figure 4a**). This is mainly because of the use of multiple different patterns in “Genome-on-Diet” for calculating the extracted seeds (as we exemplify in **Figure 3**). Even when tolerating up to two-thirds of the bases of a read (using a pattern '100'), “Genome-on-Diet” still provides distinguishable seed matching rates, while “Spaced” provides almost no distinguishability. For example, “All Seeds”, “Minimizers”, “Spaced”, and “Genome-on-Diet” provide that 18%, 24%, 78%, and 19%, respectively, of the input sequence pairs have the same or smaller seed matching rate to that of a sequence pair with an edit distance of 295. (2) For any seed matching rate, “Genome-on-Diet” has a wider range of edit distance values compared to that of both, “All Seeds” and “Minimizers” (**Figure 4a**). This means that “Genome-on-Diet” tolerates more differences than “All Seeds” and “Minimizers”, which are fundamentally designed for finding only

exact-matching seeds. This helps to find more sequence pairs with closely similar edit distance values. (3) “Genome-on-Diet” provides a similar trend to that of “All Seeds” and “Minimizers” such that with the gradual increase in the edit distance threshold, “Genome-on-Diet” accepts *gradually* more sequence pairs based on their seed matching rates (**Figure 4b**). “Genome-on-Diet” accepts slightly more sequence pairs than both “All Seeds” and “Minimizers”, which is in line with the second observation. As the number of zeros in the pattern sequence increases, “Spaced” becomes *ineffective* because it provides very high seed matching rates for both low-edit and high-edit sequence pairs. For example, “All Seeds”, “Minimizers”, “Spaced”, and “Genome-on-Diet” allow for accepting 34%, 39%, 98%, and 47%, respectively, of input sequence pairs when considering an edit distance threshold of only 138 and a pattern sequence of ‘100’. In this evaluation, we observe that none of the four seeding algorithms, “All Seeds”, “Minimizers”, “Spaced”, and “Genome-on-Diet”, provide false negatives, i.e., none of them reject a sequence pair whose edit distance is less than or equal to the target edit distance threshold.

Figure 4: (a) Seed matching rates over edit distance and (b) Number of accepted sequence pairs over different edit distance thresholds using four different seeding algorithms (all overlapping seeds, minimizer seeds, spaced seeds, and Genome-on-Diet seeds) and using four different pattern sequences ('110', '10', '101001', and '100').

Comment 5: Major point 4: In Chapter 2.1.3. the authors conclude that “Genome-on-Diet is [...] of high read mapping quality and accuracy”. First of all, accuracy was not measured as accuracy is defined as the ratio between all TPs + TNs and the total number of all true and false cases (and TN values are not defined/provided). Even if the authors define “accuracy as sensitivity” (see Chapter 1, page 4, paragraph (3)) this is misleading, in particular in this case as this suggests a low FP rate with the already shown low FN rate. Therefore I would suggest not to redefine the usual and known designations for those statistics. In addition, the authors should also, at least, include the fact that for this part of their evaluation the FP-rate for SNPs was always higher for Genome-on-Diet compared to minimap2 (for reference see Figure 2 J, K, L). In addition, this aspect could also be more discussed by the authors, i.e. the ever

present tradeoff between precision and recall and what of those Genome-on-Diet is trying to maximize and why?

Response 5: We thank the reviewer for this comment, which entails three suggestions. We make the following changes to address the three suggestions:

- 1) We replace the term “accuracy” with “detection of a higher number of true genomic variations and a lower number of missed variations” to make our claim more precise and clear any confusion. This was already clear in the abstract, but we make it now consistent everywhere in the paper, in Section 2.2.3 (it was Section 2.1.3 in the first submission) as highlighted below in yellow. This addresses the first part of the reviewer’s comment.

We conclude that Genome-on-Diet is very fast, memory-frugal, and its read mapping results lead to the detection of a higher number of true genomic variations and a lower number of missed variations compared to the state-of-the-art read mapper, minimap2.

- 2) We add a new paragraph to the observation list in Section 2.2.3 (as highlighted in yellow below) to highlight the increased number of incorrectly detected variations provided by Genome-on-Diet. This addresses the second part of the reviewer’s comment.

(3) Similar to what we observe when evaluating different pattern sequences (Section 2.2.2), Genome-on-Diet still comes with the drawback of increasing the number of incorrectly detected variations of all types.

- 3) We add a new paragraph in Section 2.2.2 to discuss the increased number of incorrectly detected variants. This addresses the last part of the reviewer’s comment.

(3) Regardless of the pattern used, sparsified read mapping comes with the drawback of increasing the number of incorrectly detected variations. This is because of the ability of Genome-on-Diet to tolerate more differences between seeds compared to other methods (e.g., minimizer seeds) that use non-sparsified (contiguous) seeds. This leads to detecting more mapping locations per read and hence more variations (both true and false) to be detected. This can be possibly addressed by applying quality filtering mechanisms that, for example, examine the number of matching bases within the matching seeds instead of only quantifying the matching seeds (as in our location voting step) since the seeds are sparsified.

Comment 6: Major point 5: At the end of Chapter 2.2.2. the authors conclude that Genome-on-Diet is maximally beneficial for very large-scale analysis with resource-limited infrastructure. This, again, is somewhat exaggerated. First of all, very large-scale genomics becomes interesting in the range of terabytes not gigabytes as used here. So the question would be to what extent Genome-on-Diet could tone down e.g. the memory requirement if using the complete RefSeq database, or at least about 10-25%, while also providing meaningful results? Secondly, the authors argue that the RefSeq database approximately doubles every 3 years (page 12, Chapter 2.2.), which corresponds to an exponential increase. However, their improvement on speed and memory preservation is clearly shown to be linear, i.e. using the pattern “110” (that is essentially two third of the input data) in comparison to “100” (that is one third of the input data) results in a linear improvement (one third to two third). How is Genome-on-Diet proposed to counter the exponential data explosion then?

Response 6: We apologize for the exaggerated claim. We change every occurrence of “very large database” or “very large-scale” into “large database” or “large-scale”, respectively, in the abstract,

conclusion, and Section 2.3.2. We also removed the following sentence: “This makes sparsified genomics ideal for truly population-scale genomic analyses and on-site analyses where compute infrastructure is usually with limited resources.”

For the scalability concern, we do agree with the reviewer that the benefits provided by Genome-on-Diet scale linearly with the number of zeros determined in the pattern, but Genome-on-Diet provides the additional key benefit of building the index on-the-fly without requiring a large peak memory footprint. The state-of-the-art tools, KMC3+CMash and Kraken2, that perform containment search produce an index of RefSeq genomes (3.5 TB) equal to 24.5 TB (7 x 3.5 TB) and 7 TB (2 x 3.5TB), respectively. While building the index using these state-of-the-art tools takes much longer than that of Genome-on-Diet as we demonstrate, loading and querying such a huge index from storage into the main memory is another concern for the state-of-the-art tools. For example, Kraken2 still requires loading the entire index into the main memory (<https://github.com/DerrickWood/kraken2/wiki/Manual#system-requirements>).

We added the following paragraph to the end of Section 2.3.2 to discuss the reviewer's concern:

Though the benefits provided by Genome-on-Diet scale linearly with the number of zeros determined in the pattern, Genome-on-Diet provides the additional key benefit of building the index on-the-fly without requiring a large peak memory footprint. The state-of-the-art tools, KMC3+CMash and Kraken2, that perform containment search produce an index of RefSeq genomes (3.5 TB) equal to 24.5 TB (7 x 3.5 TB) and 7 TB (2 x 3.5TB), respectively. While building the index using these state-of-the-art tools takes much longer than that of Genome-on-Diet as we demonstrate, loading and querying such a huge index from storage into the main memory is another concern for the state-of-the-art tools. Challenges with TB-scale databases need to be evaluated in future work.

Comment 7: Major point 6: For their evaluation on taxonomic profiling in Chapter 2.3.1. the authors describe that they have disabled the recovery mode for this part. To our understanding this feature is, however, one of the algorithmic specifications that is responsible for sustaining a high sensitivity. Controversy, disabling this feature could result in an increased FN-rate, or if all algorithms perform equally well on this regard anyway (see Chapter 2.3.1.), enabling this feature could result in an increased FP-rate. Is this also a reason for the differences on the L1-norm-error and to what extent does this influence the overall performance?

Response 7: We thank the reviewer for highlighting this comment. The mentioned Chapter 2.3.1 by the reviewer is now Section 2.4.1 in the updated manuscript. The recovery mode should not be used for taxonomic profiling. Our newly-introduced recovery mode aims to recover the poorly mapped reads in case they satisfy certain conditions such as the number of matching seeds between the read and the reference genome. This is helpful for read mapping since 1) it is usually assumed that both the sequencing reads and the reference genome belong to the same organism, and 2) the state-of-the-art read mapper minimap2 reports such poorly mapped reads. However, enabling the recovery mode for taxonomic profiling leads to forcing (depending on whether the number of matching seeds satisfies the recovery condition) the input read to get mapped (counted) to one of the indexed reference genomes despite the high dissimilarity between the read sequence and such genome. To avoid the wrong calculation of relative abundance, we decided to disable the recovery mode for taxonomic profiling.

To make this clear, we modify the subject paragraph into the following:

We disable the recovery mode in Genome-on-Diet for taxonomic profiling as the recovery mode forces each read sequence to map to one of the reference genomes stored in the reference database. This

affects the profiling results and especially the relative abundance calculation as it depends on the number of mapped reads to each taxa. Thus, Genome-on-Diet for taxonomic profiling considers only reads with a sufficient number of matching seeds for further analyses.

For the non-zero L1-norm-error, we believe the differences are mainly due to the ability of Genome-on-Diet to tolerate more differences when matching seeds. This leads to obtaining more mapping locations (in a single or multiple reference genomes) per read as we discuss in the new section (Section 2.1.2). The relative abundance calculation is directly affected by the number of uniquely mapped reads and multimapped reads and hence Genome-on-Diet provides different values for the relative abundance. For more information on how the relative abundance is calculated, we refer the reviewer to Section “Alignment and profile generation” in <https://genomebiology.biomedcentral.com/articles/10.1186/s13059-020-02159-0>

To address this comment we added the following paragraph to Section 2.4.4:

The differences are mainly due to the ability of Genome-on-Diet to tolerate more differences when matching seeds. This leads to obtaining more mapping locations (in one or more reference genomes) per read (Section 2.1.2), which directly affects the relative abundance calculations.

Comment 8: Major point 7: In Chapter 2.3.1 (first paragraph) the authors argue that the higher memory footprint of Genome-on-Diet in comparison to Metalign can be counteracted by reducing the batch size. However, taking the results shown in Table 2, a reduced batch size comes at the cost of an increased run time. This conflicts with the overall goal and claim that Genome-on-Diet is superior in terms of both memory requirements and runtime.

Response 8: We agree with the reviewer that Table 2 (Table 4 in the updated manuscript) shows a reduced batch size that comes at the cost of an increased run time. However, the total execution time of Genome-on-Diet is still low compared to that of Metalign as we added a new paragraph with an example to Section 2.4.4:

The batch size can be reduced such that Genome-on-Diet provides a comparable memory footprint to that of Metalign (Table 4) at the cost of a slight increase in execution time. For example, using a batch size of 10 billion bases reduces the peak memory footprint from 35.17 GB to only 18.59 GB while increasing the total execution time of taxonomic profiling from 8'697 seconds to 12'850 seconds (using CAMI Low), which is still 36.6x (470'885/12'850) faster than that of Metalign.

Comment 9: Minor point 1: In the PAF format output by the tool, the query's starting location on the reference is 1 index lower, resulting in the start to end position longer than the read length itself.

Response 9: We thank the reviewer for this comment. We observe that the PAF file format considers a 0-based coordinate system (Section “Output Format” <https://lh3.github.io/ minimap2/minimap2.html>), while SAM file format considers a 1-based coordinate system (Section 1.2 <https://samtools.github.io/hts-specs/SAMv1.pdf>). Thus, our current implementation complies with the standard PAF and SAM formats.

Here are examples of the output PAF for both Genome-on-Diet and Minimap2. Please note the third column (start location) and the fourth column (end location):

GDiet-LongReads:

Q1 70 0 70 + CCNA2_HUMAN 1299 420 490 70 70 60
 NM:i:0 ms:i:140 AS:i:140 nn:i:0 tp:A:P cm:i:0 s1:i:140 s2:i:0 de:f:0 rl:i:0
 cg:Z:70M

GDiet-ShortReads:

Q1 70 0 70 + CCNA2_HUMAN 1299 420 490 70 70 60
 NM:i:0 ms:i:140 AS:i:140 nn:i:0 tp:A:P cm:i:0 s1:i:140 s2:i:0 de:f:0 rl:i:0
 cg:Z:70M

Minimap2:

Q1 70 0 70 + CCNA2_HUMAN 1299 420 490 70 70 40
 NM:i:0 ms:i:140 AS:i:140 nn:i:0 tp:A:P cm:i:9 s1:i:64 s2:i:0 de:f:0 rl:i:0 cg:Z:70M

Comment 10: Minor point 2: The order of the Figures and Tables should match their occurrence in the main manuscript (e.g. see page for where Table 4 is referred even before Table one). I suggest to relabel or rearrange this accordingly.

Response 10: We thank the reviewer for bringing this up. The goal of this sentence (which we also copied below) on Page 4 is to back up the claim we have. We are not explaining the aforementioned figures/tables, but rather just a citation to the figures/tables. We prefer to keep it this way so that the reader can easily verify the correctness. As an alternative, we could remove the two references and just say: “as we experimentally demonstrate in the results section”, but it will be very broad.

For example, the execution time of both the indexing and seeding steps accounts for about 10%-27% of the total read mapping time (Figure 5D,E,F) and 97% of the total time for taxonomic profiling of metagenomic samples (Table 6).

Comment 11: Minor point 3: The authors write that Genome-on-Diet provides a lossless compression of genomic sequences. To my understanding this is an exaggeration or at least somewhat misleading. If the original genomic sequence needs to be retained (as only a copy of the original genomic sequence is sparsified) in order to reverse the process losslessly it is hardly a compression, even if the sequences are compressed by other means (i.e. using gzip). At any rate, compression is not the application scenario for this newly introduced method. Therefore, I argue against raising such expectations unless it is shown by a proper evaluation or proven methodologically.

Response 11: We apologize for the confusion. To clear the potential confusion, we replace all occurrences of the word “compression” as follows:

Section 1.1: The pattern sequence is a *user-defined, fully-configurable* shortest repeating substring that represents included and excluded bases via 1’s and 0’s, respectively. Genome-on-Diet manipulates *only a copy of the genomic sequence that is used for the initial steps of an analysis, such as indexing and seeding. The original genomic sequence is still maintained (by default) for performing accuracy-critical steps of an analysis, if needed, such as base-level sequence alignment where all bases must be*

accounted for. Users can disable maintaining the original genomic sequence by using the `--idx-no-seq` parameter setting.

Section 3: The primary purpose of Genome-on-Diet is to significantly reduce the end-to-end execution time and memory footprint of the indexing and seeding steps in genomic analyses by sparsifying genomic sequences and enabling processing of the sparsified, shorter genomic sequences.

Section 3: The reduction ratio of a genomic sequence is $\beta = p/x$.

Section 3.1: Genome-on-Diet alters *only* a copy of the input reference genome for building the seed index.

Comment 12: Minor point 4: In Chapter 2.2.2. (page 14) the authors explain that “Genome-On-Diet” requires carefully adjusting the thresholds for the location voting step when using different pattern sequences”. What are the recommendations here if any other than the default pattern is used? What were the precise parameters for all parts of the three different evaluations? Does this parameter also need to be adjusted for different input data or just for different patterns?

Response 12: We thank the reviewer for this comment, which entails three points. To address each, we make the following changes:

1. We removed the subject sentences (which are highlighted in yellow below) as we already performed several new experiments (Sections 2.1.2 and 2.2.2) using different pattern sequences, without changing any other parameter value and the results look logical.

Genome-on-Diet requires carefully adjusting the thresholds for the location voting step when using different pattern sequences. Genome-on-Diet conveniently allows users to configure both the pattern sequence and the voting thresholds.

2. We added the precise parameters and command lines for the four evaluations (Sections 2.1, 2.2, 2.3, and 2.4) are all provided in the GitHub repo: <https://github.com/CMU-SAFARI/Genome-on-Diet/tree/main/ReproducibleEvaluation>

3. The meaning of each parameter setting and comments on the value are all provided when running the tool as follows:

```
./GDiet_avx
```

Reviewer #2 (Remarks to the Author):

Comment 13: The manuscript by Alser et al. presents a novel strategy to deal with the continuously increasing amount of genomic data and the very mapping analysis tasks that are commonly done using this data.

The manuscript is well-written and organized, although there are a few points that should be considered and revised, according to this reviewer's opinion.

Response 13: We thank the reviewer for finding our manuscript well-written and organized.

Comment 14: One is from the title, after reading through the manuscript, the "Genomic Analyses" words in the title might be too broad. For instance, analyses such as gene calling and how will they be affected by the use of this approach. Is in this reviewer's opinion that this approach can be very useful, as presented in the manuscript, for mapping tasks, but other genomic analyses might not be performed the same as when using the full genomic data. One suggestion for revising the title could be "Taming Large-Scale Mapping Analyses via Sparsified Genomics", but others the authors might think will better give a grasp of their approach could work as well.

Response 14: We thank the reviewer for this comment. We agree with the reviewer that the applicability of Genome-on-Diet is more clear for applications that rely on mapping analyses. We briefly mention in the discussion section as quoted below several genomic analyses that can benefit from sparsified genomics. While some of these analyses have indexing or mapping steps involved, others do not have a mapping step per se such as pre-alignment filtering where we compare a sequence with another sequence. For this reason, we would like to keep the title as is since we already reasoned about the broad applicability as follows:

"Our work has broad applicability as we demonstrate benefits in read mapping, containment search, and robust microbiome discovery. In addition, other potential applications are pangenome mapping^{3,120}, quantification of transcript expression¹²¹, identifying *de novo* variations by directly comparing read sequences between related individuals⁶², identifying somatic variations by directly comparing reads sequenced from normal and tumor genomes¹⁸, and pre-alignment filtering¹¹⁰."

Comment 15: Another aspect that is stressed throughout the manuscript is the reduced size of the database by employing this approach. This is in general true if you only consider the database, but to be honest the full genomic data must be stored and present somewhere (which might not be true for other mappers for which the full sequence can be directly reconstructed from the indexed database). Hence, this should be made clear up-front and maybe not stressed that much. As if someone will need to change the binary pattern would still need to go back to the full genomic data.

Response 15: We thank the reviewer for this comment. Our argument about reducing the size of the database is always correct as the reviewer stated and as we experimentally demonstrate with the three applications. The approach to store a copy of the complete genome sequence in the index is a user-configurable parameter, which is enabled by default. To disable storing the copy of the genome sequence, the user can add this flag to the command line of Genome-on-Diet: --idx-no-seq. This approach is already adopted in the state-of-the-art read mapper, minimap2, and state-of-the-art

metagenomic profiler, Metalign, and hence it is up to the user to keep the reference genome stored in the index. To address this comment, we added the following sentence:

Genome-on-Diet manipulates *only* a copy of the genomic sequence that is used for the initial steps of an analysis, such as indexing and seeding. The original genomic sequence is still maintained (by default) for performing accuracy-critical steps of an analysis, if needed, such as base-level sequence alignment where all bases must be accounted for. Users can disable maintaining the original genomic sequence by using the `--idx-no-seq` parameter setting.

Comment 16: A third important aspect is probably not stressed enough. It is understandable that it is difficult to collect and present all aspects in a very detailed manner. However, the choice of the binary pattern (or 'diet pattern') seems to be very crucial for having comparable performances when using the Genome-on-Diet framework. So, practically, to make this easily adopted by the community, it will be very important to be able to provide some examples about the effect of different choices for this pattern and how this can also affect performances.

Response 16: We thank the reviewer for this important suggestion. We agree with the reviewer on the need for evaluating the effect of using different patterns. We now added several new experiments where we evaluated 12 different pattern sequences for short read mapping and 3 more pattern sequences for long read mapping. We describe the new experiments and the resulting observations in Section 2.2.2 as shown below. We also clarify and exemplify the key differences between spaced seeding and Genome-on-Diet using different pattern sequences as shown in Figure 3, Figure 4, and the new Section 2.1.

2.2.2. Sparsifying Genomic Sequences Significantly Accelerates Read Mapping

The first question we need to answer is which pattern sequence a user can choose for sparsified read mapping. To answer this key question we evaluate the effect of using different pattern sequences on the detected indels and SNPs and the execution time of Genome-on-Diet (**Table 2**). We make five key observations: (1) Compared to non-sparsified read mapping (using Genome-on-Diet with a pattern of '11'), sparsified read mapping can increase the number of correctly detected genomic variations as sparsified read mapping allows tolerating more differences when performing seed matching. (2) The use of pattern '10' provides the best performance when considering all evaluated metrics collectively compared to all evaluated pattern sequences, which is expected based on our analysis (**Section 2.1.2**). The use of pattern '10' increases the number of correctly detected variations by 4% and decreases both the number of missed variations and the execution time of read mapping by 25.9% and 28.4%, respectively, compared to that provided by the non-sparsified read mapping. (3) Regardless of the pattern used, sparsified read mapping comes with the drawback of increasing the number of incorrectly detected variations. This is because of the ability of Genome-on-Diet to tolerate more differences between seeds compared to other methods (e.g., minimizer seeds) that use non-sparsified (contiguous) seeds. This leads to detecting more mapping locations per read and hence more variations (both true and false) to be detected. This can be possibly addressed by applying quality filtering mechanisms that, for example, examine the number of matching bases within the matching seeds instead of only quantifying the matching seeds (as in our location voting step) since the seeds are sparsified. (4) The location of zeros in the pattern sequence has a slight effect on all evaluated metrics. That is, the patterns '110', '101', and '011' (similarly patterns '101001', '100101', '001101') all provide a similar performance.

However, the number of zeros in the pattern sequence has a significant effect on all evaluated metrics. (5) Though both '10' and '101001' patterns lead to excluding half of the bases from the reference genome and read sequences, they lead to different read mapping performances. This is mainly because each pattern may result in applying different patterns to each seed depending on the location of the extracted seed in the diet pattern. For example, the pattern '10' results in applying one of the two patterns, '101010...' or '01010...', to each overlapping seed, while the pattern '101001' results in applying one of the three patterns, '1010011...', '01001101...', or '001101001...', to each overlapping seed (Figure 3c). This directly affects the number of potential mapping locations that need to be examined, which in turn can affect the mapping results.

We conclude that the use of pattern '10' still provides the best accuracy-speedup tradeoffs for sparsified short read mapping. We also make the same observation for long read mapping (Table 3). Thus, we decide to use the pattern '10' for the read mapping application.

Table 2: Number of (correctly detected, incorrectly detected, and missed) indels/SNPs and total execution time as provided by Genome-on-Diet using different pattern sequences and Illumina preset (k=21 and w=11).

Pattern Sequence	INDELS / SNPs			Execution time (sec)
	Correctly detected	Incorrectly detected	Missed	
11	18'362	86	2'862	12'814.20
110	19'099	963	2'125	11'129.16
101	19'023	1'066	2'201	11'116.24
011	19'070	881	2'154	11'034.62
10	19'105	787	2'119	9'178.78
1001	18'982	1'370	2'242	9'130.61
100	19'025	1'462	2'199	8'086.17
101001	18'067	2'531	3'157	6'351.91
100101	18'194	2'463	3'030	6'565.39
001101	18'087	2'736	3'137	6'499.64
1111100000	18'840	1'763	2'384	9'926.99
1001010011	18'811	1'740	2'413	9'936.54

Table 3: Number of (correctly detected, incorrectly detected, and missed) indels/SNPs and SVs along with the total execution time as provided by Genome-on-Diet using different pattern sequences and HiFi preset (k=19 and w=19).

Pattern Sequence	INDELS / SNPs	SVs	Execution time (sec)
------------------	---------------	-----	----------------------

	Correctly detected	Incorrectly detected	Missed	Correctly detected	Incorrectly detected	Missed	
11	16'731	1'964	4'493	180	11	36	42'494.66
10	17'795	2'028	3'429	193	57	23	37'787.72
100	14'042	1'822	7'182	141	57	75	15'080.8

2.1. Sparsified Genomics is a Unique Novel Mechanism

To our knowledge, this work is the *first* to introduce the concept of sparsifying genomic sequences and processing sparsified sequences in a very fast, efficient, and accurate way. Genome-on-Diet is *fundamentally different* from other techniques (e.g., spaced seeds^{46–51}) that apply patterns to genomic sequences in five important aspects. (1) Genome-on-Diet applies a repeating pattern to the genomic sequence (i.e., reference genome or sequencing read), while spaced seeding keeps the genomic sequence unchanged and applies a pattern to each extracted seed (**Figure 3**). (2) The resulting seed in Genome-on-Diet spans a much wider region in the reference genome compared to spaced seeds, which is of vital importance for containment search and metagenomics applications. (3) Genome-on-Diet can use a pattern sequence of any length, while spaced seeding has to use a pattern sequence of length equal to the seed length. (4) Genome-on-Diet avoids extracting seeds that start at a location overlapping with a corresponding 0 in the pattern sequence, while spaced seeding extracts the same number of seeds regardless of the pattern used. For example, using a pattern of '1000101001' results in extracting *only* 3 overlapping seeds in Genome-on-Diet, while spaced seeding extracts 6 overlapping seeds from the same region (**Figure 3**). (5) The resulting seeds in Genome-on-Diet are eventually formed based on different patterns depending on the pattern sequence and the location of the seed with respect to the repeating pattern, while spaced seeding usually applies the same pattern to all seeds. For example, with a pattern of 1000101001, Genome-on-Diet forms the first three seeds with patterns of '1000101001', '0001010011', and '0100110001', respectively, while spaced seeding forms all seeds with a pattern of '1000101001' (**Figure 3**). We also provide other examples of how a single user-defined pattern leads to applying different patterns to each seed (**Figure 3c**). We experimentally investigate the benefits and downsides of the differences between Genome-on-Diet and spaced seeding.

Figure 3: Example of resulting seeds after applying (a) spaced seeding and (b) Genome-on-Diet on a given genome sequence using the same user-defined pattern. (c) Examples of how each overlapping seed may have its own pattern. The examples at the top, middle, and bottom use user-defined patterns of '10', '101', and '101001', respectively.

2.1.1. Evaluation Methodology

Unfortunately, we are not aware of any recent read mapper that uses spaced seeding and can compete with the state-of-the-art read mapper, minimap2. To be fair with spaced seeding, we build a computer program that 1) takes a genomic sequence, 2) introduces a copy of the input sequence with a random number of substitutions at random locations such that we evaluate sequence pairs with a wide range of locations and number of differences, 3) extracts genomic seeds from the input sequence and stores them in a list, 4) extracts genomic seeds from the mutated sequence and quantify how many of these seeds appear in the list of stored seeds, and 5) repeats the first four steps for every input genomic sequence. Our computer program uses four different algorithms to extract seeds: all overlapping seeds (“All Seeds” in **Figure 4**), minimizer seeds (“Minimizers”), spaced seeds (“Spaced”), and Genome-on-Diet seeds (“Genome-on-Diet”). We use “All Seeds” and “Minimizers” algorithms as a reference for the number of matching seeds. It outputs the Levenshtein distance⁷¹ for each sequence pair along with the seed matching rates calculated when using each of the four seeding algorithms. We empirically run our computer program using a seed length of 8 and a minimizer window of size 6 and using 1000-long sequences. We use four different pattern sequences, '110', '10', '101001', and '100'. We make our computer program publicly available through the same GitHub project of this work.

2.1.2. Sparsified Genomics Allows For Tolerating Variations

We evaluate the effect of using multiple different patterns for extracted seeds as in Genome-on-Diet compared to using the same pattern sequence for all extracted seeds as in spaced seeding (**Figure 4**). We use two performance metrics: 1) *Seed matching rate*, which we define as the ratio of the number of matching seeds (i.e., seeds extracted from one sequence that match the seeds extracted from the other sequence) to the total number of extracted seeds. Ideally, the higher the seed matching rate between two sequences the higher the similarity between the two sequences. Thus, a well-performing seeding algorithm should provide an inversely proportional relationship between the seed matching rate and edit distance. 2) *Number of accepted sequence pairs*, which we define as the number of sequence pairs that have a seed matching rate greater than or equal to a seed matching rate threshold. We determine the seed matching rate threshold as the minimum seed matching rate for all sequence pairs that have an edit distance less than or equal to a specific edit distance threshold. For example, If we want to accept all sequence pairs that have an edit distance threshold of 20, we calculate both the edit distance and the seed matching rate for all input sequence pairs and we consider the minimum seed matching rate of all sequence pairs that have an edit distance of at most 20 as the target seed matching rate threshold.

We make two observations. (1) “Genome-on-Diet” shows higher distinguishability than “Spaced” between sequence pairs with low edit distance and sequence pairs with high edit distance (**Figure 4a**). This is mainly because of the use of multiple different patterns in “Genome-on-Diet” for calculating the extracted seeds (as we exemplify in **Figure 3**). Even when tolerating up to two-thirds of the bases of a read (using a pattern '100'), “Genome-on-Diet” still provides distinguishable seed matching rates, while

“Spaced” provides almost no distinguishability. For example, “All Seeds”, “Minimizers”, “Spaced”, and “Genome-on-Diet” provide that 18%, 24%, 78%, and 19%, respectively, of the input sequence pairs have the same or smaller seed matching rate to that of a sequence pair with an edit distance of 295. (2) For any seed matching rate, “Genome-on-Diet” has a wider range of edit distance values compared to that of both, “All Seeds” and “Minimizers” (Figure 4a). This means that “Genome-on-Diet” tolerates more differences than “All Seeds” and “Minimizers”, which are fundamentally designed for finding only exact-matching seeds. This helps to find more sequence pairs with closely similar edit distance values. (3) “Genome-on-Diet” provides a similar trend to that of “All Seeds” and “Minimizers” such that with the gradual increase in the edit distance threshold, “Genome-on-Diet” accepts *gradually* more sequence pairs based on their seed matching rates (Figure 4b). “Genome-on-Diet” accepts slightly more sequence pairs than both “All Seeds” and “Minimizers”, which is in line with the second observation. As the number of zeros in the pattern sequence increases, “Spaced” becomes *ineffective* because it provides very high seed matching rates for both low-edit and high-edit sequence pairs. For example, “All Seeds”, “Minimizers”, “Spaced”, and “Genome-on-Diet” allow for accepting 34%, 39%, 98%, and 47%, respectively, of input sequence pairs when considering an edit distance threshold of only 138 and a pattern sequence of ‘100’. In this evaluation, we observe that none of the four seeding algorithms, “All Seeds”, “Minimizers”, “Spaced”, and “Genome-on-Diet”, provide false negatives, i.e., none of them reject a sequence pair whose edit distance is less than or equal to the target edit distance threshold.

Figure 4: (a) Seed matching rates over edit distance and (b) Number of accepted sequence pairs over different edit distance thresholds using four different seeding algorithms (all overlapping seeds, minimizer seeds, spaced seeds, and Genome-on-Diet seeds) and using four different pattern sequences ('110', '10', '101001', and '100').

Comment 17: Lastly, a minor point, but since bowtie2 is also a very common mapping tool, it will be very nice by the authors if they could include it as well in their comparisons.

Response 17: We thank the reviewer for this comment. We perform a new experiment for evaluating Bowtie2 using its two best configurations: very fast mapping and very sensitive mapping. We describe the experiment and the results as follows:

In the evaluation Methodology Section:

We also evaluate the performance of Bowtie2⁸⁴ using its latest version (2.5.1). Bowtie2 requires building an index separately before performing read mapping. We run Bowtie2 in two mapping modes: very fast mapping (--very-fast) with a small index (default) and very sensitive mapping (--very-sensitive) with a large index (--large-index).

In the observation list:

(5) Genome-on-Diet is 5.24x and 17.26x faster than Bowtie2⁸⁴ in “very fast” mapping mode and “very sensitive” mapping mode, respectively, (**Supplementary Table 3**). Bowtie2 provides 1.56-2.1x lower peak memory footprint (as Bowtie2 uses FM-index, which results in a compressed, small index based on the Burrows–Wheeler transformation) and 4.2-5.7x higher storage footprint than Genome-on-Diet.

In the read mapping quality Section:

This is also true for Genome-on-Diet over Bowtie2 (**Supplementary Table 3**).

In supplementary results:

Supplementary Table 3. Number of (correctly detected, incorrectly detected, and missed) indels/SNPs, total execution time, peak memory footprint, and storage footprint provided by Genome-on-Diet, minimap2, and Bowtie2 using Illumina presets. We use two best configurations for Bowtie2: very fast mapping with a small index and very sensitive mapping with a large index.

Pattern		11	10	minimap2	Bowtie2 (very fast, small index)	Bowtie2 (very sensitive, large index)
INDELS / SNPs	Correctly detected	18'362	19'105	18'178	17'803	17'664
	Incorrectly detected	86	787	14	56	46
	Missed	2'862	2'119	3'046	3'421	3'560
Execution time (sec)		12'814.2	9'178.78	31'358.75	74'613.77	344'820.58
Peak memory footprint (GB)		13.8	8.615	14.376	4.14	5.52
Storage footprint (GB)		0	0	0	4.2	5.7

Reviewer #2 (Remarks to the Author):

Comment 1: This reviewer thanks the authors for the revised version of the manuscript. All points were addressed and the newly added information and comparison improved the overall quality and accessibility of the manuscript.

Response 1: We thank the reviewer for 1) approving our changes and responses to the reviewer's concerns and 2) confirming that our changes in response to the reviewer's comments improved the overall quality and accessibility of the manuscript.

Reviewer #3 (Remarks to the Author):

Comment 1: The authors appropriately addressed most of the comments raised in the previous review round, also providing additional material.

Response 1: We thank the reviewer for acknowledging that we appropriately addressed the concerns raised by the two reviewers during the first round.

Comment 2: However, in my opinion, among the newly provided material, the results presented in section "2.1.1 Evaluation methodology" and "2.1.2 Sparsified genomics allow for tolerant variations" are not completely fair. I list below the main concerns: The patterns chosen for Genome-on-Diet are among those that best perform (according to the results shown in subsequent sections). However, it is not clear how the parameters were chosen for the other pattern schemes to which they are compared. For all-seed (which I understand are basically k-mers) a length 8 is chosen. Why is it so? Did the authors tried several lengths and showed the best results? If so this should be made clear in the manuscript. The same holds for minimizers: size 6 in a window of length 8 is the best performing combination for minimizers or a randomly chose one?

Response 2: We thank the reviewer for highlighting the incomplete information. We chose a seed length of 8 based on the "Seedability" paper [1] (Table 1 in their paper). The paper claims a better minimap2 performance when using a seed length between 5 and 10 for 1000-long sequences. We chose a minimizer window size of 6 based on the recommendation of minimap2's manual reference (<https://lh3.github.io/minimap2/minimap2.html>) to use a window size that equals 2/3rds of the seed length.

[1] Ayad, Lorraine AK, Rayan Chikhi, and Solon P. Pissis. "Seedability: optimizing alignment parameters for sensitive sequence comparison." *Bioinformatics Advances* 3, no. 1 (2023): vbad108.

To address the reviewer's concern, we now evaluate the four seeding algorithms using four different seed lengths, 8, 13, 18, and 21, instead of a single seed length. The three new seed lengths, 13, 18, and 21, are within the range of seed length used practically for minimap2. We perform the following three changes:

1) We add the following clarification to Section 2.1.1.

We run our computer program using four different seed lengths (k), 8, 13, 18, and 21, and a minimizer window of size (w) 6, 9, 12, and 11, respectively, using 1000-long sequences. The seed length of 8 is suggested in⁷², while the other three seed lengths are within the practical range of seed length used for minimap2³⁷. The first three window sizes are calculated as 2/3rds of the seed length based on the reference manual of minimap2⁷³. The last minimizer window is calculated based on the default preset of minimap2, `-x sr`. We use four pattern sequences (P), '110', '10', '101001', and '100' for a seed length of 8 to examine the effect of using different ratios of the number of zeros to the number of ones. For each of the other seed lengths (13, 18, and 21), we use the pattern sequence '10' and another spaced-seeding-friendly pattern sequence suggested by the literature. These patterns are

'1110110110111'⁷⁴ for a seed length of 13, '111001011001010111'⁷⁵ for a seed length of 18, and '11110110110101110111'⁷⁴ for a seed length of 21. We make our computer program publicly available through the same GitHub project of this work.

2) We update Section 2.1.2 to reflect the new evaluation results for these seed lengths.

2.1.2. Sparsified Genomics Allows For Tolerating Variations

We evaluate the four different seeding algorithms, all overlapping seeds, minimizer seeds, spaced seeds, and Genome-on-Diet seeds (**Figure 4**). We use two performance metrics: 1) *Seed matching rate*, which we define as the ratio of the number of matching seeds (i.e., seeds extracted from one sequence that match the seeds extracted from the other sequence) to the total number of extracted seeds. Ideally, the higher the seed matching rate between two sequences the higher the similarity between the two sequences. Thus, a well-performing seeding algorithm should provide an inversely proportional relationship between the seed matching rate and edit distance. 2) *Number of accepted sequence pairs*, which we define as the number of sequence pairs that have a seed matching rate greater than or equal to a seed matching rate threshold. We determine the seed matching rate threshold as the minimum seed matching rate for all sequence pairs that have an edit distance less than or equal to a specific edit distance threshold. For example, If we want to accept all sequence pairs that have an edit distance threshold of 20, we calculate both the edit distance and the seed matching rate for all input sequence pairs, and we consider the minimum seed matching rate of all sequence pairs that have an edit distance of at most 20 as the target seed matching rate threshold.

We make four observations. (1) Using a seed length of 8, "Genome-on-Diet" shows higher distinguishability than "Spaced" between sequence pairs with low edit distance and sequence pairs with high edit distance (**Figure 4a**). This is mainly because of the use of multiple different patterns in "Genome-on-Diet" for calculating the extracted seeds (as we exemplify in **Figure 3**). Even when tolerating up to two-thirds of the bases of a read (using a pattern '100'), "Genome-on-Diet" still provides distinguishable seed matching rates, while "Spaced" allows most of the compared seeds to be highly similar to each other providing almost no distinguishability (proven theoretically⁷⁶). For example, "All Seeds", "Minimizers", "Spaced", and "Genome-on-Diet" provide that 18%, 24%, 78%, and 19%, respectively, of the input sequence pairs have the same or smaller seed matching rate to that of a sequence pair with an edit distance of 295. (2) Using a seed length of 8, "Genome-on-Diet" accepts slightly more sequence pairs than both "All Seeds" and "Minimizers" (**Figure 4b**). As the number of zeros in the pattern sequence increases, "Spaced" becomes *ineffective* because it tolerates more differences between compared seeds allowing for very high seed matching rates for both low-edit and high-edit sequence pairs (proven theoretically⁷⁶). For example, "All Seeds", "Minimizers", "Spaced", and "Genome-on-Diet" allow for accepting 34%, 39%, 98%, and 47%, respectively, of input sequence pairs when considering an edit distance threshold of only 138 and a pattern sequence of '100'.

(3) "Genome-on-Diet" still provides high distinguishability between low-edit and high-edit sequence pairs even when using longer seeds and different pattern sequences (**Figure 4c**). Using both longer seed length and spaced-seeding-friendly pattern significantly improve the distinguishability of "Spaced". (4) For any seed matching rate, "Genome-on-Diet" has a wider range of edit distance values compared to that of "All Seeds", "Minimizers", and "Spaced" (**Figure 4c**). This means that "Genome-on-Diet" tolerates more differences than "All Seeds", "Minimizers", and "Spaced". This helps to find more sequence pairs with closely similar edit distance values. This is clearly demonstrated when the

number of zeros in the pattern sequence increases (a pattern sequence of '10') (**Figure 4d**). In our evaluation, we observe that none of the four seeding algorithms, "All Seeds", "Minimizers", "Spaced", and "Genome-on-Diet", provide false negatives, i.e., none of them reject a sequence pair whose edit distance is less than or equal to the target edit distance threshold.

3) We add two new figures, Figure 4(c) and Figure 4(d), to Section 2.1.2.

Figure 4: Performance of four seeding algorithms, all overlapping seeds, minimizer seeds, spaced seeds, and Genome-on-Diet seeds. We evaluate seed matching rates over edit distance when using a seed length of (a) 8 and (c) 13, 18, and 21. We quantify the number of accepted sequence pairs over different edit distance thresholds when using a seed length of (b) 8 and (d) 13, 18, and 21.

Comment 3: As for spaced-seeds I understand the authors wanted to test the same 0-1 patterns used for genome-on-diet, but this is not fair because, as the authors explained themselves, spaced-seeds work in a very different manner, much closer to how k-mers work (but including of course wildcards). I will detailed several specific observations about spaced-seeds in the following comments. In all the literature that I am aware of, spaced-seeds are defined as patterns of 0s and 1s that always begin and end with a 1. According to this definition, 3 out of 4 of the tested patterns are not technically spaced-seeds. I point out to the authors, as a valuable source of information about spaced-seeds, the web page:

<https://sites.google.com/view/laurentnoe/spaced-seeds>

Although I see it is not longer maintained, the literature list is updated up to 2020, so it has all the references related to basics definitions. In real applications the positions of the 1s in a spaced-seeds mask is not random, but the result of the maximization (or minimization) of an objective function (e.g. minimizing the overlap or maximizing the sensitivity). There are tools that can compute the best distribution of 1s and 0s, e.g. rasbhari:

L. Hahn, C.-A. Leimeister, R. Ounit, S. Lonardi, and B. Morgenstern, "rasbhari: optimizing spaced seeds for database searching, read mapping and alignment-free sequence comparison," PLoS Computational Biology, vol. 12, p. e1005107, October 2016.

Other examples of "good" spaced-seeds can be found in the paper related to the target application of interests.

Response 3: We thank the reviewer for this feedback. We agree with the reviewer on the importance of choosing a "good" pattern sequence for spaced seeding. To address the reviewer's concern, we use spaced-seeding-friendly pattern sequences suggested by the literature for each evaluated seed length. We obtained the spaced-seeding-friendly pattern sequences from the following two papers, which are highlighted in the webpage (<https://sites.google.com/view/laurentnoe/spaced-seeds>) suggested by the reviewer:

[1] Mallik, A. and Ilie, L., 2021. ALeS: adaptive-length spaced-seed design. *Bioinformatics*, 37(9), pp.1206-1210.

[2] Noé, L., 2017. Best hits of 11110110111: model-free selection and parameter-free sensitivity calculation of spaced seeds. *Algorithms for Molecular Biology*, 12, pp.1-16.

We did not use the *rasbhari* paper suggested by the reviewer as *rasbhari* has not been maintained since 2017 (<http://rasbhari.gobics.de>).

To address the reviewer's concern, we changed Section 2.1.1 and Section 2.1.2 as we highlight in our Response #2 above to describe the newly added pattern sequences and present the new evaluation results.

Comment 4: The length of the spaced-seed mask that was tested is limited to 2 or 3 nucleotides for 3 out of 4 of the tested spaced-seed masks. This length is definitely too short for sequences of length 1000. If $k=8$ is indeed the best length for the all-seed scheme, then I guess reasonable combination of length-weight for spaced-seeds (where with weight I mean the number of 1s) could be 8-6 or 10-8. In any case, this is just a guess and proper experiment should be done. Since for genome-on-diet the best pattern were shown, also some test with different parameters for the other schemes should be carried out.

Response 4: We thank the reviewer for this feedback. The spaced-seed masks we used for a seed length of 8 were 110, 10, 10100, and 100. These pattern sequences correspond to the following 8-long pattern sequences: 11011011, 10101010, 10100101, 10010010. Hence, we believe these sequences are reasonable and represent different ratios of the number of zeros to the number of ones.

To address the reviewer's concern, we use three more seed lengths, 13, 18, and 21. For a seed length of 13, we use the patterns '1010101010101' and '1110110110111'. For a seed length of 18, we use the patterns '101010101010101010' and '111001011001010111'. For a seed length of 21, we use the patterns '101010101010101010101' and '111101101101011101111'. We change Section 2.1.1 and Section 2.1.2 as we highlight in our Response #2 above to describe the newly added pattern sequences and present the new evaluation results.